# Bayesian Domain Adaptation with Gaussian Mixture Domain-Indexing

**Yanfang Ling**
Sun Yat-sen University
lingyf3@mail2.sysu.edu.cn

**Jiyong Li**
Sun Yat-sen University
lijy373@mail2.sysu.edu.cn

**Lingbo Li**
InfMind Technology Ltd
lingbo@infmind.ai

**Shangsong Liang**[*]
Sun Yat-sen University
liangshangsong@gmail.com

## Abstract

Recent methods are proposed to improve performance of domain adaptation by inferring domain index under an adversarial variational bayesian framework, where domain index is unavailable. However, existing methods typically assume that the global domain indices are sampled from a vanilla gaussian prior, overlooking the inherent structures among different domains. To address this challenge, we propose a Bayesian Domain Adaptation with **G**aussian **M**ixture **D**omain-**I**ndexing(GMDI) algorithm. GMDI employs a Gaussian Mixture Model for domain indices, with the number of component distributions in the "*domain-themes*" space adaptively determined by a Chinese Restaurant Process. By dynamically adjusting the mixtures at the domain indices level, GMDI significantly improves domain adaptation performance. Our theoretical analysis demonstrates that GMDI achieves a more stringent evidence lower bound, closer to the log-likelihood. For classification, GMDI outperforms all approaches, and surpasses the state-of-the-art method, VDI, by up to 3.4%, reaching 99.3%. For regression, GMDI reduces MSE by up to 21% (from 3.160 to 2.493), achieving the lowest errors among all methods. Source code is publicly available from `https://github.com/lingyf3/GMDI`.

## 1 Introduction

Machine learning models often suffer from performance degradation when applied to new domains that differ from their training domains, a phenomenon known as domain shift [21, 8, 28, 15]. Domain Adaptation (DA) [4, 44, 49, 42, 6, 1, 37, 32, 38, 13] seeks to mitigate this issue by producing domain-invariant features, thereby enhancing generalization from source to target domains [23, 19, 12, 39, 45].

Recent research has explored the use of domain identity and domain index to improve domain-invariant data encoding and enhance domain adaptation performance [36, 40, 41]. *Domain identity* [41], a one-hot discrete variable vector, differentiates between domains, whereas *domain index* [41], a real-valued continuous variable vector, captures domain semantics. Due to the limited information in the discrete domain identity vector, research has increasingly focused on the domain index. Current approaches to incorporating domain index in domain adaptation include: (1) Directly using existing additional information in the dataset as the domain index [36, 40], which is impractical for datasets lacking such indices [22, 29], and (2) Treating the domain index as a latent variable to be inferred [26, 41]. However, these methods typically

---

[*]Corresponding author.

38th Conference on Neural Information Processing Systems (NeurIPS 2024).

model the domain indices with a simple Gaussian distribution, limiting the domain indices space and thus hindering adaptation to diverse target domains, resulting in suboptimal performance.

To address the aforementioned issues, we propose a Bayesian Domain Adaptation with Gaussian Mixture Domain-Indexing (GMDI) algorithm. The proposed adversarial Bayesian algorithm assumes that domain indices follow a mixture of Gaussian distributions, with the number of mixture components dynamically determined by a Chinese Restaurant Process. As shown in Figure 1, a single Gaussian distribution struggles to adequately fit the domain indices, neglecting the inherent structures among different domains. This observation motivates us to model domain indices from different domains collectively as a Gaussian mixture distribution. To the best of our knowledge, we are the first to model domain indices as a mixture of Gaussian distributions to address the aforementioned challenges. Inspired by [3], the latent space of the mixture is defined as the "domain-themes" space.

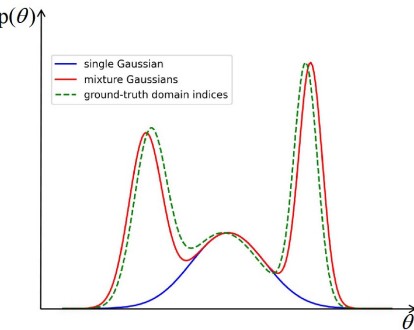

Figure 1: Illustration of domain indices modeled by different distributions.

The mixtures of distributions provide a higher level of flexibility in a larger latent space, thereby increasing the capability to adapt to various target domains with domain shift. Our theoretical analysis demonstrates that GMDI achieves a more rigorous evidence lower bound, and that maximizing this bound along with adversarial loss effectively infers optimal domain indices. Extensive experimental results validate the significant effectiveness of GMDI.

Our key contributions are summarized as: (1) Our proposed GMDI is the first one to consider the entire distributions of domain indices in the "domain-themes" space following a mixture of Gaussian distributions, and dynamically determining the number of components in the mixture with the Chinese Restaurant Process. (2) Our detailed theoretical analysis demonstrates that training with GMDI's superior evidence lower bound together with adversarial loss can yield optimal and more interpretable domain indices. (3) Extensive experiments on classification and regression tasks showcase the strong domain index modeling capability of GMDI, significantly outperforming the state-of-the-art.

## 2   Related Work

**Adversarial domain adaptation.**  There exists a substantial body of work on domain adaptation [4, 44, 49, 42, 6, 1, 37, 32, 38]. They focus on generating domain-invariant data encoding by aligning the distributions of source and target domains to adapt to target domains. This alignment is achieved by directly matching the statistics of distributions [25, 24] or by employing adversarial loss [33, 31], which encourages domain confusion through adversarial objective with a discriminator. Adversarial domain adaptation is widely used due to its integration with deep learning, strong theoretical foundation [7], and superior performance. Various different types of adversarial losses have been explored: [35] uses an inverted label GAN loss, [5] utilizes a minimax loss, and [34] employs a cross-entropy loss against the uniform distribution. Typically, the discriminators in these models rely on *domain identity*, which contains limited information, to align data encoding distributions. [20] and [10] also pay attention to domain identity. Our work, however, focuses on *domain index*, providing a more detailed representation of domains.

**Domain adaptation related to domain indices.** Recently, there has been growing interest in using continuous *domain index*, which contain richer and more interpretable information, to enhance domain adaptation performance. [36] use the rotation angle of images as the domain index for the Rotating MNIST dataset and patients' ages as the domain index for Healthcare Datasets. Their theoretical analysis demonstrates the value of utilizing domain indices to generate domain-invariant features. [40] employ graph node embeddings as domain indices to achieve domain adaptation in graph-relational domains. These methods assume that domain indices are available. However, in practice, domain indices are not always accessible [22, 29]. [26] generates features representing the similarity between different domains but do not formally define the domain index. [41] formally define the domain index and treat it as a latent variable to be inferred. Although [41] takes steps towards Bayesian approximation to parameter distributions, it only assumes a single domain index

distribution, limiting its capability to adapt to diverse target domains effectively. In contrast, we address this issue by representing the domain index with a dynamically updated mixture model.

## 3 Background

### 3.1 Problem setup

We aim at unsupervised domain adaptation: given $N$ domains with different domain shifts, each domain has a domain identity $w \in \mathcal{W} = [N] \triangleq \{1, ..., N\}$, and each domain contains $D_w$ data points. Similar to the conventional unsupervised domain adaptation setting, the $N$ domains are divided into source domains with labeled data $\mathcal{D}^S = \{(\boldsymbol{x}_i^s, y_i^s, w_i^s)\}_{i=1}^{n_s}$ and target domains with unlabeled data $\mathcal{D}^T = \{(\boldsymbol{x}_i^t, w_i^t)\}_{i=1}^{n_t}$. A foundational element that builds up our research problem is the diverse domain shifts [14] between different target domains and source domains. For source domains, the complexity of each target domain varies, which motivates us to dynamically infer domain indices in the "domain-themes" space and model them with dynamic Gaussian Mixture Model. We aim to (1) predict the label $\{y_i^t\}_{i=1}^{n_t}$ of target domain data, and (2) infer local domain index $\boldsymbol{u}_w \in \mathbb{R}^{B_u}$ and global domain index $\boldsymbol{\theta}_w \in \mathbb{R}^{B_\theta}$ in the dynamic "domain-themes" space. The summary of the notations is presented in Appendix J.

### 3.2 Preliminary

**Domain index.** The domain index, distinct from domain identity $w$, represents domain semantics, thus empowering it to significantly enhance domain adaptation performance. As per its definition [36, 41], the domain index satisfies the following : (1) To acquire domain-invariant data encoding $\boldsymbol{z}$, the global domain index $\boldsymbol{\theta}$ must remain independent of data encoding $\boldsymbol{z}$ , i.e., $\boldsymbol{\theta} \perp\!\!\!\perp \boldsymbol{z}$ or equivalently $p(\boldsymbol{z} \mid \boldsymbol{\theta}) = p(\boldsymbol{z})$. (2) Effectively representing data point $\boldsymbol{x}$ while averting the occurrence of collapsing. (3) Ensuring optimal performance of downstream tasks utilizing the data encoding $\boldsymbol{z}$ learned by the encoder under the aforementioned constraints, and necessitating the maintenance of sensitivity to labels.

**Variational domain index.** In circumstances where the domain index may not be readily accessible, the Variational Domain Index (VDI) [41] is a Bayesian approach to infer the domain index $\boldsymbol{\theta}$ and $\boldsymbol{u}$ as latent variables. VDI factorizes the generative model $p(\boldsymbol{x}, y, \boldsymbol{u}, \boldsymbol{\theta}, \boldsymbol{z} \mid \boldsymbol{\varepsilon})$ as:

$$p(\boldsymbol{x}, y, \boldsymbol{u}, \boldsymbol{\theta}, \boldsymbol{z} \mid \boldsymbol{\varepsilon}) = p(\boldsymbol{\theta} \mid \boldsymbol{\varepsilon})p(\boldsymbol{u} \mid \boldsymbol{\theta})p(\boldsymbol{x} \mid \boldsymbol{u})p(\boldsymbol{z} \mid \boldsymbol{x}, \boldsymbol{u}, \boldsymbol{\theta})p(y \mid \boldsymbol{z}), \quad (1)$$

where $\boldsymbol{\varepsilon}$ denotes the parameters for the prior probability distribution of the domain index $\boldsymbol{\theta}$. As shown in Equation 1, VDI stands capable of significantly enhancing domain adaptation proficiency by leveraging the inferred domain index for generating data encoding $\boldsymbol{z}$. Note that the independence between the domain index $\boldsymbol{\theta}$ and data encoding $\boldsymbol{z}$, i.e., $p(\boldsymbol{z} \mid \boldsymbol{\theta}) = p(\boldsymbol{z})$, does not contradict $p(\boldsymbol{z} \mid \boldsymbol{x}, \boldsymbol{u}, \boldsymbol{\theta})$, given the existence of multiple pathways between the domain index $\boldsymbol{\theta}$ and data encoding $\boldsymbol{z}$. Compared to VDI, which treats the distribution of domain index as a single Gaussian, we focus on a dynamic mixture of Gaussian distributions.

**Chinese Restaurant Process(CRP).** The Dirichlet Process (DP) is a classical method used for clustering. However, DP is difficult to construct directly, we apply the Chinese Restaurant Process(CRP) [16, 17, 18, 46] to implement it. Since similar domains have similar domain indices, the clusters formed by the domain indices correspond one-to-one with the components in mixture of Gaussian distributions. CRP can be employed to determine which cluster a domain belongs to (i.e., which component distribution domain index corresponds to). Specially, CRP is able to dynamically and adaptively determine the number of mixture components. CRP operates as follows:

$$P(v = k) = \begin{cases} \dfrac{n_k}{N - 1 + \alpha} & \text{if the cluster } k \text{ exists}, \\ \dfrac{\alpha}{N - 1 + \alpha} & \text{if cluster } k \text{ is a new cluster}, \end{cases} \quad (2)$$

where $n_k$ is the number of domain contained in cluster $k$, and parameter $\alpha$ is the concentration parameter of the CRP. A larger $\alpha$ implies a tendency to generate more domain clusters.

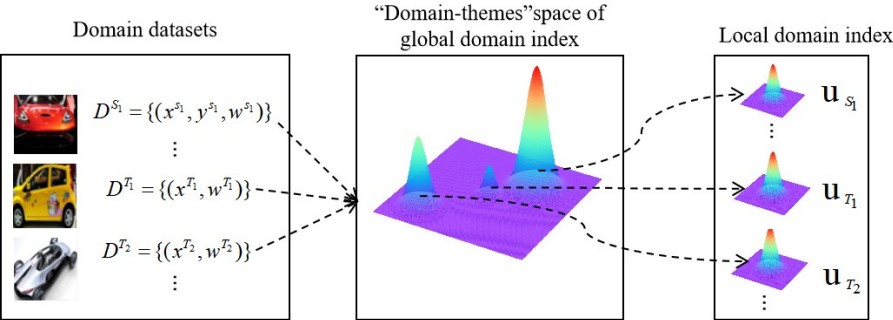

Figure 2: The schematic diagram of domain index distributions. It shows the inference of variational Gaussian-shaped distributions for the global domain index, representing domain semantics. The process involves ranking candidate distributions in the "domain-themes" space, selecting the highest probability one, and deriving the local domain index from it.

# 4 Bayesian Domain Adaptation with Gaussian Mixture Domain-Indexing

## 4.1 Overview of GMDI

We propose GMDI in order to infer more interpretable domain indices and thereby improve domain adaptation performance. Our model is constructed in three steps: First, in generate process, we model global domain indices as a dynamic Gaussian mixture model, with local indices generated from global domain index. Second, in inference process, we build structured variational inference to approximate the posterior of the latent variables. Finally, we train the model using an evidence lower bound with robust theoretical guarantees and an adversarial loss. Under this framework, GMDI has several significant advantages: (1) With global domain indices following a dynamic mixture of Gaussian distributions adaptively determined by CRP, it provides a higher level of flexibility in a larger latent space. (2) The evidence lower bound of our GMDI is more stringent, leading to more interpretable and optimal domain indices. The overview of GMDI is presented in Algorithm 1.

## 4.2 Mixture of domain index distributions

Similar to VDI, GMDI (Figure 3 (right)) also considers the intermediate latent variable of local domain index $\boldsymbol{u}$. The local domain index $\boldsymbol{u}$ contains instance-level information, meaning that each data point has a unique local domain index. In contrast, the global domain index $\boldsymbol{\theta}$ contains domain-level information, indicating that all data points within the same domain share the same global domain index. In VDI, with the local domain index $\boldsymbol{u}$ derived from the global domain index $\boldsymbol{\theta}$, the data distribution $p(y, \boldsymbol{x} \mid \boldsymbol{\varepsilon})$ is expressed as:

$$p(y, \boldsymbol{x} \mid \boldsymbol{\varepsilon}) = \int p(\boldsymbol{\theta} \mid \boldsymbol{\varepsilon}) p(\boldsymbol{u} \mid \boldsymbol{\theta}) p(\boldsymbol{x} \mid \boldsymbol{u}) p(\boldsymbol{z} \mid \boldsymbol{x}, \boldsymbol{u}, \boldsymbol{\theta}) p(y \mid \boldsymbol{z}) \, d\boldsymbol{z} d\boldsymbol{u} d\boldsymbol{\theta} \,, \tag{3}$$

where $\boldsymbol{\varepsilon}$ denotes the parameters for the prior probability distribution of the global domain index $\boldsymbol{\theta}$.

With the setting of unsupervised domain adaptation, domains are not i.i.d, existing domain shift between domains. It implies that there may be substantial differences in distribution between domains. This leads to a problem that disparate target domains require a more significant degree of adaptation. Although we compute a distribution for domain index $\boldsymbol{\theta}$ to enhance the capability of domain adaptation, it may not be effective enough to aid in adapting to a diversity of different target domains. Therefore, if local domain index $\boldsymbol{u}$ are adapted from a simple Gaussian distribution of global domain index $\boldsymbol{\theta}$, it may play a small role in improving the performance of domain adaptation.

To better model global domain index and thus enhance the effectiveness of domain adaptation, we propose to maintain a mixture of dynamically updated global domain index $\boldsymbol{\theta}$ distributions in the "domain-themes" space. Intuitively, similar domains have similar global domain indices, implying that the mixture of global domain index distributions is associated with a cluster of similar domains. The process of adapting local domain indices from global domain indices is illustrated in Figure 2. Specifically, we consider a Gaussian Mixture Model (GMM) as the mixture of global domain index

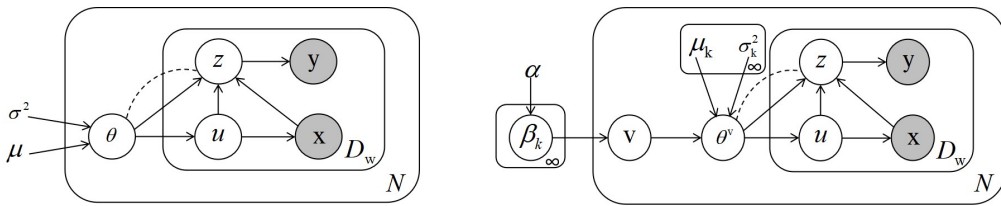

Figure 3: The probabilistic graphical model of VDI (**left**) and GMDI (**right**). Edge type "- - -" denotes the independence between global domain index $\boldsymbol{\theta}$ and data encoding $\boldsymbol{z}$.

distributions. For each distinct domain, we first rank the candidate global domain index distributions in "domain-themes" space and select the distribution with the highest probability. We then derive the local domain index $\boldsymbol{u}$ from the global domain index $\boldsymbol{\theta}$. Therefore, the global domain index is designed to be dynamical GMM.

Let $v$ denote the latent categorical variable indicating the assignment of a domain to a cluster, which is equivalent to selecting components in a mixture distribution. Based on the definition of $v$, we derive the updated representation of the distribution $p(y, \boldsymbol{x} \mid \boldsymbol{\varepsilon})$:

$$p(y, \boldsymbol{x} \mid \boldsymbol{\varepsilon}) = \int p(v)p(\boldsymbol{\theta}^v \mid \boldsymbol{\varepsilon})p(\boldsymbol{u} \mid \boldsymbol{\theta}^v)p(\boldsymbol{x} \mid \boldsymbol{u})p(\boldsymbol{z} \mid \boldsymbol{x}, \boldsymbol{u}, \boldsymbol{\theta}^v)p(y \mid \boldsymbol{z}) \, d\boldsymbol{z}d\boldsymbol{u}d\boldsymbol{\theta}dv \,. \tag{4}$$

The component distribution $\boldsymbol{\theta}^v$ is selected from the mixture distribution of $\boldsymbol{\theta}$, and afterward, the local domain index $\boldsymbol{u}$ is obtained from global domain index $\boldsymbol{\theta}^v$ for generating domain-invariant data encoding $\boldsymbol{z}$. Equation 3 and Equation 4 both represent the factorization of the distribution $p(y, \boldsymbol{x} \mid \boldsymbol{\varepsilon})$. $\boldsymbol{\theta}$ in Equation 3 is the global domain index. While GMDI models the global domain index $\boldsymbol{\theta}$ as a mixture of Gaussian distributions, $\boldsymbol{\theta}^v$ in Equation 4 indicates the $v$-th component of the mixture distribution of $\boldsymbol{\theta}$ with the prior $p(v)$. Compared to the single distribution, the mixture of global domain index distributions adequately model the domain index of different domains, enhancing the effectiveness of domain adaptation in the face of varying degrees or even significant domain shifts. However, a remaining challenge is determining the number of components in the mixture distributions, especially when there are numerous domains, or even possibly infinite ones.

### 4.3 Generative process of GMDI

In extreme cases, there may be infinite domains. Due to CRP's flexibility in dynamically determining the number of domain indices mixture components, we employ CRP to determine which cluster a domain belongs to (i.e., which component distribution domain index corresponds to). Specifically, we define the prior for domain indices cluster as a CRP, where the generation of new domain indices clusters is controlled by parameter $\alpha$. Thus, the probability of a domain belonging to cluster $k$ is calculated by Equation 2. Since CRP is an infinite mixture model, it is able to easily adapted to an infinite number of domains.

Mixture of domain indices need a stick-breaking representation of CRP to obtain component weights. Stick-breaking representation indicating an infinite construction, considering the Dirichlet prior with parameter $\alpha$, each element in the probability vector $\boldsymbol{\pi} = [\pi_1, \pi_2, ...]$ is non-negative and the sum of the elements is 1:

$$\beta_k \mid \alpha \sim \text{Beta}(1, \alpha) \text{ for } k\text{=1, ...,}\infty \,, \tag{5}$$

$$\pi_k = \beta_k \prod_{l=1}^{k-1} (1 - \beta_l) \text{ for } k\text{=1, ...,}\infty \,. \tag{6}$$

Equation 6 is equivalent to the weights implied by CRP. Based on Equation 6 and Equation 4, the generative process of GMDI is as follows:

$$v \mid \boldsymbol{\pi} \sim \text{Categorical}_\infty(\boldsymbol{\pi}) \,, \tag{7}$$

$$\boldsymbol{\theta}^{v=k} \sim \mathcal{N}(\boldsymbol{\mu}_k, \boldsymbol{\sigma}_k^2) \,, \tag{8}$$

$$\boldsymbol{u} \mid \boldsymbol{\theta}^{v=k} \sim p(\boldsymbol{u} \mid \boldsymbol{\theta}^{v=k}) \,, \tag{9}$$

$$\boldsymbol{x} \mid \boldsymbol{u} \sim p(\boldsymbol{x} \mid \boldsymbol{u}) \,, \tag{10}$$

$$\boldsymbol{z} \mid \boldsymbol{x}, \boldsymbol{u}, \boldsymbol{\theta}^v \sim p(\boldsymbol{z} \mid \boldsymbol{x}, \boldsymbol{u}, \boldsymbol{\theta}^v) \,, \tag{11}$$

where $\boldsymbol{\mu}_k$ and $\boldsymbol{\sigma}_k^2$ are mean vector and semi-positive covariance matrix of the $k$-th component in dynamic Gaussian mixture of domain indices, respectively. Figure 3 illustrates the generative process of VDI with a single distribution and GMDI with a mixture of distributions for domain indices. Since CRP is computationally intensive. To improve computational efficiency, we consider the stick-breaking construction to transform the infinite Gaussian mixture of domain indices into a finite one. It can be achieved by directly specifying an upper bound $K$ for the number of components in Gaussian mixture of domain indices. Selecting an appropriate $K$ allows to effectively reduce computational overhead. The finite version of the generative process of GMDI is available in Appendix A.

Accordingly, the generative model can be factorized as follows:

$$p(\boldsymbol{x}, y, \boldsymbol{u}, \boldsymbol{\theta}, \boldsymbol{z}, v, \boldsymbol{\beta} \mid \alpha) = p(\boldsymbol{\beta} \mid \alpha)p(v \mid \boldsymbol{\beta})p(\boldsymbol{\theta}^v)p(\boldsymbol{u} \mid \boldsymbol{\theta}^v)p(\boldsymbol{x} \mid \boldsymbol{u})p(\boldsymbol{z} \mid \boldsymbol{x}, \boldsymbol{u}, \boldsymbol{\theta}^v)p(y \mid \boldsymbol{z}). \quad (12)$$

The predictor $p(y \mid \boldsymbol{z})$ is a categorical distribution for classification tasks and a Gaussian distribution for regression tasks.

## 4.4 Evidence Lower Bound

The exact posterior of all latent variables, i.e., $p(\boldsymbol{u}, \boldsymbol{\theta}, \boldsymbol{z}, v, \boldsymbol{\beta} \mid \boldsymbol{x})$ is intractable, variational inference is used to approximate the posterior. Compared to the Monte Carlo sampling, variational inference allows both uncertainty quantification and computational efficiency. We employ structured variational inference to approximate the exact posterior, factorizing the approximate posterior $q(\boldsymbol{u}, \boldsymbol{\theta}, \boldsymbol{z}, v, \boldsymbol{\beta} \mid \boldsymbol{x})$:

$$q(\boldsymbol{u}, \boldsymbol{\theta}, \boldsymbol{z}, v, \boldsymbol{\beta} \mid \boldsymbol{x}) = q(\boldsymbol{\beta}; \boldsymbol{\gamma})q(v; \boldsymbol{\eta})q(\boldsymbol{u} \mid \boldsymbol{x}; \boldsymbol{\psi}_u)q(\boldsymbol{\theta}^v \mid \boldsymbol{u}; \boldsymbol{\psi}_\theta)q(\boldsymbol{z} \mid \boldsymbol{x}, \boldsymbol{u}, \boldsymbol{\theta}^v; \boldsymbol{\psi}_z), \quad (13)$$

where $\boldsymbol{\gamma}, \boldsymbol{\eta}, \boldsymbol{\psi}_u, \boldsymbol{\psi}_\theta$ and $\boldsymbol{\psi}_z$ respectively represent the parameters of the variational distributions $q(\boldsymbol{\beta}), q(v), q(\boldsymbol{u} \mid \boldsymbol{x}), q(\boldsymbol{\theta}^v \mid \boldsymbol{u})$ and $q(\boldsymbol{z} \mid \boldsymbol{x}, \boldsymbol{u}, \boldsymbol{\theta}^v)$.

We train GMDI by maximizing the evidence lower bound(ELBO) to obtain the optimal variational distributions which best approximate exact posterior distributions. Section 5 demonstrates that our proposed GMDI has a more stringent evidence lower bound. Based on generative and inference process of GMDI, we caluculate the ELBO as follows:

$$
\begin{aligned}
\mathcal{L}_{\mathrm{ELBO}} = {} & \mathbb{E}_{q(\boldsymbol{u}, \boldsymbol{\theta}^v, \boldsymbol{z} \mid \boldsymbol{x}; \boldsymbol{\phi})q(v; \boldsymbol{\eta})}[\log p(y|\boldsymbol{z})] + \mathbb{E}_{q(\boldsymbol{u}|\boldsymbol{x}; \boldsymbol{\psi}_u)}[\log p(\boldsymbol{x}|\boldsymbol{u})] \\
& + \mathbb{E}_{q(v; \boldsymbol{\eta})q(\boldsymbol{\beta}; \boldsymbol{\gamma})q(\boldsymbol{u}|\boldsymbol{x}; \boldsymbol{\psi}_u)q(\boldsymbol{\theta}^v|\boldsymbol{u}; \boldsymbol{\psi}_\theta)}[\log p(\boldsymbol{u}|\boldsymbol{\theta}^v)] - \mathrm{KL}[q(\boldsymbol{\beta}; \boldsymbol{\gamma})||p(\boldsymbol{\beta})] \\
& - \mathbb{E}_{q(\boldsymbol{\beta}; \boldsymbol{\gamma})}[\mathrm{KL}[q(v; \boldsymbol{\eta})||p(v|\boldsymbol{\beta}; \boldsymbol{\psi}_v)]] - \mathbb{E}_{q(\boldsymbol{u}|\boldsymbol{x}; \boldsymbol{\psi}_u)q(v; \boldsymbol{\eta})}[\mathrm{KL}[q(\boldsymbol{\theta}^v|\boldsymbol{u}; \boldsymbol{\psi}_\theta)||p(\boldsymbol{\theta}^v)]] \\
& - \mathbb{E}_{q(v; \boldsymbol{\eta})q(\boldsymbol{u}, \boldsymbol{\theta}^v|\boldsymbol{x}; \boldsymbol{\xi})}[\mathrm{KL}[q(\boldsymbol{z}|\boldsymbol{x}, \boldsymbol{u}, \boldsymbol{\theta}^v; \boldsymbol{\psi}_z)||p(\boldsymbol{z}|\boldsymbol{x}, \boldsymbol{u}, \boldsymbol{\theta}^v)]] \\
& - \mathbb{E}_{q(\boldsymbol{u}|\boldsymbol{x}; \boldsymbol{\psi}_u)}[\log q(\boldsymbol{u}|\boldsymbol{x}; \boldsymbol{\psi}_u)],
\end{aligned}
\quad (14)
$$

where $\boldsymbol{\phi}$ and $\boldsymbol{\xi}$ represent the parameters of the variational distributions $q(\boldsymbol{u}, \boldsymbol{\theta}^v, \boldsymbol{z}|\boldsymbol{x})$ and $q(\boldsymbol{u}, \boldsymbol{\theta}^v|\boldsymbol{x})$, respectively, and $\mathrm{KL}[\cdot||\cdot]$ is the Kullback–Leibler divergence.

## 4.5 Adversarial loss with a discriminator

To ensure the independence between global domain index $\boldsymbol{\theta}$ and data encoding $\boldsymbol{z}$ as defined, we follow VDI [41] by training an additional discriminator D with an adversarial loss. As we prove in Section 5 that the independence between global domain index $\boldsymbol{\theta}$ and data encoding $\boldsymbol{z}$ relies on the independence between domain identity $w$ and data encoding $\boldsymbol{z}$, the adversarial loss is simplified to discriminate the domain identity $w$:

$$\mathcal{L}_{\mathrm{D}} = \mathbb{E}_{p(w, \boldsymbol{x})}\mathbb{E}_{q(\boldsymbol{z}|\boldsymbol{x}; \boldsymbol{\psi}_z)}[\log \mathrm{D}(w|\boldsymbol{z})]. \quad (15)$$

## 4.6 Objective function

Combining Equation 14 and Equation 15, the final objective of GMDI is:

$$\mathcal{L}_{\mathrm{GMDI}} = \max \min_{D} \mathcal{L}_{\mathrm{ELBO}} - \lambda * \mathcal{L}_{\mathrm{D}}, \quad (16)$$

where $\lambda$ denotes the hyper-parameter that balances two terms. Since the exact posterior of all latent variables is intractable, we propose to use a structured variational inference method to approximate the exact posterior. More details can be viewed in the appendix B.

**Variational distribution of $\boldsymbol{\beta}$.** To derive the optimal variational distribution of $\boldsymbol{\beta}$, we only consider the terms related to $\boldsymbol{\beta}$ in $\mathcal{L}_{\text{ELBO}}$, then we can get the posterior $q(\boldsymbol{\beta}_k; \boldsymbol{\gamma}_k) = \text{Beta}(\boldsymbol{\beta}_k; \gamma_{k,1}, \gamma_{k,2})$ with parameters $\gamma_{k,1} = 1 + \boldsymbol{\eta}_k$ and $\gamma_{k,2} = \alpha + \sum_{i=k+1}^{K} \boldsymbol{\eta}_i$.

**Variational distribution of $v$.** Similarly, the variational posterior of $v$ can be calculated as a Categorical distribution $q(v; \boldsymbol{\eta}) = \text{Categorical}_K(v; \boldsymbol{\eta})$, where the pareameters can be updated as:

$$
\begin{aligned}
\log \boldsymbol{\eta}_k \propto & \mathbb{E}_{q(\boldsymbol{\beta};\boldsymbol{\gamma})}[\boldsymbol{\pi}] + \mathbb{E}_{q(\boldsymbol{u}|\boldsymbol{x};\boldsymbol{\psi}_u)q(\boldsymbol{\theta}^v|\boldsymbol{u};\boldsymbol{\psi}_\theta)}[\log p(\boldsymbol{u}|\boldsymbol{\theta}^v)] - \mathbb{E}_{q(\boldsymbol{u}|\boldsymbol{x};\boldsymbol{\psi}_u)}[\text{KL}[q(\boldsymbol{\theta}^v|\boldsymbol{u};\boldsymbol{\psi}_\theta)||p(\boldsymbol{\theta}^v)]] \\
& - \mathbb{E}_{q(\boldsymbol{u},\boldsymbol{\theta}^v|\boldsymbol{x};\boldsymbol{\xi})}[\text{KL}[q(\boldsymbol{z}|\boldsymbol{u},\boldsymbol{\theta},\boldsymbol{x};\boldsymbol{\psi}_z)||p(\boldsymbol{z}|\boldsymbol{u},\boldsymbol{\theta},\boldsymbol{x})]] ,
\end{aligned}
\tag{17}
$$

where $\sum_{k=1}^{K} \boldsymbol{\eta}_k = 1$ and $q(\boldsymbol{\beta}; \boldsymbol{\gamma}) = \prod_{k=1}^{K-1} q(\boldsymbol{\beta}_k; \boldsymbol{\gamma}_k)$.

**Variational distribution of $\boldsymbol{\theta}, \boldsymbol{u}$ and $\boldsymbol{z}$.** With assuming that the latent parameters are sampled from Gaussian, we have the following forms:

$$
q(\boldsymbol{\theta}^v|\boldsymbol{u}; \boldsymbol{\psi}_\theta) = \mathcal{N}(\boldsymbol{\mu}_\theta, \boldsymbol{\sigma}_\theta^2) ,
\tag{18}
$$

$$
q(\boldsymbol{u}|\boldsymbol{x}; \boldsymbol{\psi}_u) = \mathcal{N}(\boldsymbol{\mu}_u, \boldsymbol{\sigma}_u^2) ,
\tag{19}
$$

$$
q(\boldsymbol{z}|\boldsymbol{x}, \boldsymbol{u}, \boldsymbol{\theta}^v; \boldsymbol{\psi}_z) = \mathcal{N}(\boldsymbol{\mu}_z, \boldsymbol{\sigma}_z^2) ,
\tag{20}
$$

where $\boldsymbol{\mu}$ and $\boldsymbol{\sigma}^2$ are mean vector and semi-positive covariance matrix of Gaussian distribution. The parameters are updated by gradient descend. Specifically, we follow VDI [41] by using Earth Mover's Distance (EMD)[30] and Multi-Dimensional Scaling (MDS)[2] to infer $\boldsymbol{\theta}^v$ from $\boldsymbol{u}$.

# 5 Theory

In this section, we provide significant theoretical guarantees for our GMDI method. First, we give the upper bound for ELBO and adversial loss respectively. Second, we prove the upper bound of the whole loss with mutual information and entropy only. Moreover, we show that the upper bound can be achieved when the conditions are satisfied. Finally, we proove the significant result that our ELBO is better than the VDI, which means that a mixture of Gaussian prior can get better results. See Appendix C for detailed proof.

**Lemma 1** *The ELBO of $p(\boldsymbol{x}, y)$ is bounded by the following formula with the Mutual Information, the Entropy and the* KL*-divergence:*

$$
\begin{aligned}
\mathbb{E}_{p(\boldsymbol{x},y)}[\mathcal{L}_{\text{ELBO}}(p(\boldsymbol{x},y))] \leq & I(y; \boldsymbol{z}) + I(\boldsymbol{x}; \boldsymbol{u}, \boldsymbol{\theta}, \boldsymbol{z}, v) - (H(\boldsymbol{x}) + H(y)) \\
& - \mathbb{E}_{q(\boldsymbol{x},\boldsymbol{u},\boldsymbol{\theta},\boldsymbol{z},v)}[\text{KL}[q(\boldsymbol{x}|\boldsymbol{u},\boldsymbol{\theta},v,\boldsymbol{z})||p(\boldsymbol{x}|\boldsymbol{u},\boldsymbol{\theta},v,\boldsymbol{z})]] \\
& - \text{KL}[q(\boldsymbol{u},\boldsymbol{\theta},v,\boldsymbol{z}|\boldsymbol{x})||p(\boldsymbol{u},\boldsymbol{\theta},v,\boldsymbol{z})] .
\end{aligned}
$$

The main difference between and Lemma 1 in GMDI and Lemma 4.1 in VDI [41] is the last two KL terms and the inclusion of $v$.

**Lemma 2** *(Information Decomposition of the Adversarial Loss [41])We can decompose the global maximum of adversial loss as follows:*

$$
\max_D \mathbb{E}_{p(w,\boldsymbol{x})}\mathbb{E}_{q(\boldsymbol{z}|\boldsymbol{x})}[\log \text{D}(w|\boldsymbol{z})] = I(\boldsymbol{z}; \boldsymbol{\theta}) + I(\boldsymbol{z}; w|\boldsymbol{\theta}) - H(w) .
$$

*The global minimum of the function is achieved if and only if $I(\boldsymbol{z}; \boldsymbol{\theta}) = 0$ and $I(\boldsymbol{z}; w|\boldsymbol{\theta}) = 0$.*

**Theorem 1** *The upper bound of the objective function can be decomposed as follows:*

$$
\mathcal{L}_{\text{GMDI}} \leq I(y; \boldsymbol{z}) + I(\boldsymbol{x}; \boldsymbol{u}, \boldsymbol{\theta}, \boldsymbol{z}, v) - I(\boldsymbol{z}; \boldsymbol{\theta}) - I(\boldsymbol{z}; w|\boldsymbol{\theta}) - (H(\boldsymbol{x}) + H(y) - H(w)) .
$$

The main difference between Theorem 1 in GMDI and Theorem 4.1 in VDI [41] is the inclusion of $v$.

**Theorem 2** *The global optimum is achieved if and only if: (1)$I(\boldsymbol{z}; \boldsymbol{\theta}) = I(\boldsymbol{z}; w|\boldsymbol{\theta}) = 0$, (2)$I(y; \boldsymbol{z})$ and $I(\boldsymbol{x}; \boldsymbol{u}, \boldsymbol{\theta}, \boldsymbol{z}, v)$ are maximized, (3)$\text{KL}[q(\boldsymbol{u}, \boldsymbol{\theta}, v, \boldsymbol{z}|\boldsymbol{x})||p(\boldsymbol{u}, \boldsymbol{\theta}, v, \boldsymbol{z})] = 0$ and $\text{KL}[q(\boldsymbol{x}|\boldsymbol{u}, \boldsymbol{\theta}, v, \boldsymbol{z})||p(\boldsymbol{x}|\boldsymbol{u}, \boldsymbol{\theta}, v, \boldsymbol{z})] = 0$.*

The main difference between Theorem 2 in GMDI and Theorem 4.2 in VDI [41] is that $I(\boldsymbol{x}; \boldsymbol{u}, \boldsymbol{\theta}, \boldsymbol{z}, v)$, which includes $v$, needs to be maximized, and the two KL divergences should equal zero.

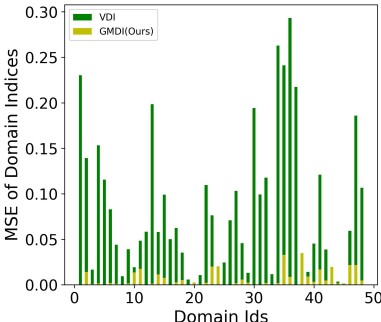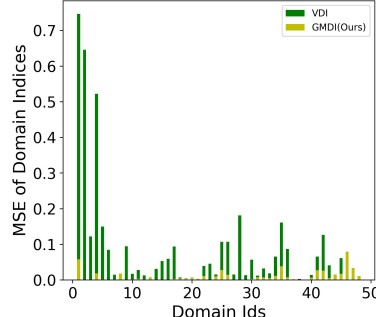

Figure 4: MSE of domain indices on *TPT-48* dataset. **Left**:N (24) → S (24), ground-truth domain indices are latitude. **Right**: W (6) → E (42), ground-truth domain indices are longitude.

Table 1: Accuracy on binary classification tasks (*Circle*, *DG-15*, and *DG-60*) and 4-way classification task (*CompCars*).

| Dataset | Method | | | | | | | | |
|---|---|---|---|---|---|---|---|---|---|
| | Source-only | DANN | ADDA | CDANN | MDD | SENTRY | D2V | VDI | GMDI (Ours) |
| *Circle* | 55.5 | 53.4 | 56.2 | 54.9 | 53.4 | 59.5 | 60.1 | 94.3 | **96.9** |
| *DG-15* | 39.7 | 43.3 | 33.5 | 38.8 | 37.2 | 42.6 | 79.9 | 94.7 | **96.5** |
| *DG-60* | 55.0 | 66.3 | 60.8 | 65.3 | 54.6 | 51.3 | 82.1 | 95.9 | **99.3** |
| *CompCars* | 39.1 | 38.9 | 42.8 | 41.8 | 41.4 | 41.8 | 40.7 | 42.5[2] | **44.4** |

**Theorem 3** *Assuming the ELBO and objective of VDI are $\mathcal{L}_{\mathrm{VDI-ELBO}}$ and $\mathcal{L}_{\mathrm{VDI}}$ respectively, where domain indices are sampled from a simple Gaussian prior, we can prove that our objective achieves a more stringent evidence lower bound which is closer to the log-likelihood, and also a tighter upper bound of the objective: $\mathcal{L}_{\mathrm{VDI-ELBO}} \leq \mathcal{L}_{\mathrm{ELBO}} \leq \log p(\boldsymbol{x}, y)$ and $\mathcal{L}_{\mathrm{VDI}} \leq \mathcal{L}_{\mathrm{GMDI}}$.*

# 6 Experimental Study

We verify the effectiveness of GMDI via experimental comparison and analysis. In particular, we answer three research questions: (**RQ1**) Can the performance of GMDI for domain adaptation outperform baselines? (**RQ2**) How effective is the global domain indices inferred by GMDI? (**RQ3**) How does the number of mixture components K affect results? Additional experimental results are available in Appendix K.

## 6.1 Experimental setup

**Datasets.** We compare GMDI with existing DA methods on the following datasets (see Appendix H and Appendix I for more details): *Circle* [36] is used for binary classification task. *DG-15* and *DG-60* [40] are synthetic datasets used for binary classification task. *TPT-48* [40] dataset is a real-world dataset used for regression task. W (6) → E (42): Adapting models from the 6 states in the west to the 42 states in the east. N (24) → S (24): Adapting models from the 24 states in the north to the 24 states in the south. *level-1 target domains*: one hop away from the closest source domain. *level-2 target domains*: two hops away from the closest source domain. *level-3 target domains*: more than two hops away from the closest source domain. *CompCars* [43] dataset is a real-world dataset for 4-way classification task.

**Baselines.** To evaluate our proposed GMDI, we compare it against eight state-of-the-art domain adaptation methods: Domain Adversarial Neural Networks (**DANN**) [4], Adversarial Discriminative Domain Adaptation (**ADDA**) [5], Conditional Domain Adaptation Neural Networks (**CDANN**) [48], Margin Disparity Discrepancy (**MDD**) [47], **SENTRY** [27], Domain to Vector (**D2V**) [26], and Variational Domain Index (**VDI**) [41]. Additionally, we include the results for models trained and tested only on the source domain (**Source-only**). Note that D2V is not applicable to regression tasks, so its results are not reported on the *TPT-48* dataset. Moreover, since our proposed GMDI focuses on inferring domain indices when they are unavailable, whereas [36] and [40] assume domain indices

---

[2]Reproduced result from VDI.

Table 2: MSE for various DA methods in both tasks W (6) → E (42) and N (24) → S (24) on *TPT-48*. We report the average MSE of all domains as well as more detailed average MSE of level-1, level-2, level-3 target domains, respectively. Note that there is only one single DA model per column. We mark the best result with **bold face**.

| Task | Domain | Source-only | DANN | ADDA | CDANN | MDD | SENTRY | VDI | GMDI(Ours) |
|------|--------|-------------|------|------|-------|-----|--------|-----|------------|
| W (6) → E (42) | Average of 4 level-1 domains | **1.184** | 1.984 | 5.448 | 6.168 | 5.544 | 2.515 | 2.160 | 1.346 |
| | Average of 6 level-2 domains | 3.128 | 5.112 | 7.624 | 7.016 | 7.912 | 5.136 | 3.000 | **2.393** |
| | Average of 32 level-3 domains | 5.272 | 5.880 | 7.256 | 6.986 | 8.008 | 5.872 | 2.448 | **2.122** |
| | Average of all 42 domains | 4.576 | 5.400 | 7.136 | 6.896 | 7.76 | 5.456 | 2.496 | **2.087** |
| N (24) → S (24) | Average of 10 level-1 domains | 1.648 | 1.832 | 5.872 | 1.832 | 2.736 | 3.976 | 1.536 | **1.479** |
| | Average of 6 level-2 domains | 3.128 | 3.296 | 6.888 | 2.856 | 6.144 | 3.760 | 2.584 | **2.119** |
| | Average of 8 level-3 domains | 9.280 | 6.744 | 7.088 | 7.688 | 10.608 | **3.672** | 5.624 | 3.942 |
| | Average of all 24 domains | 4.560 | 3.840 | 6.528 | 4.040 | 6.216 | 3.816 | 3.160 | **2.493** |

are available, they are not applicable to our setting. Detailed explanations of these algorithms can be found in the respective references.

## 6.2 Results and discussion

**RQ1: Performance on classification and regression tasks.** (1) *Circle*, *DG-15* and *DG-60*. The results in Table 1 show that, on all three datasets, the performance of baselines other than D2V and VDI is only marginally better or worse than random guess (accuracy of 50%). This is likely due to the complex relationships between domains within the datasets, making it difficult to adapt to target domains. Additionally, Source-only performs poorly due to overfitting. Compared to VDI, our GMDI improves accuracy by up to 3.4%, achieving very high accuracy (over 96.5%). This improvement is attributed to proposal of modeling the global domain index as the mixture distributions (e.g., Figure 8). (2) *TPT-48*. In Table 2, we report the mean square error (MSE) of the evaluated methods on *TPT-48*. In both E (6) → W (42) and N (24) → S (24) regression tasks, all methods except DANN, SENTRY, and VDI performed worse than Source-only, indicating the occurrence of negative transfer. In contrast, GMDI significantly reduced the MSE compared to VDI, with average MSE decreases of 16% and 21%, respectively. (3) *CompCars*. The results in Table 1 show that our method achieves the best classification accuracy. All domain adaptation methods improved to varying degrees compared to Source-only, but our method achieved the highest increase in accuracy, with an improvement of up to 5%. In Figure 7(left), the data encoding generated by VDI are clustered together, indicating a mixture of points from different class labels. In contrast, in Figure 7(right), the data encoding of GMDI are separated by class label, demonstrating that GMDI can better distinguish points by class label. **Across all datasets, GMDI significantly outperforms baselines, with minimum accuracy of 96.5% on synthetic datasets , while MSE is reduced by at least 16% on *TPT-48* dataset.**

**RQ2: Effectiveness of inferred domain indices.** Note that our proposed GMDI have no access to ground-truth domain indices $\theta$ during traning. To evaluate the effectiveness of GMDI in inferring domain indices, we compare the inferred domain indices with the ground-truth domain indices and calculate MSE between them. As shown in Figure 5, for nearly all 30 domains on *Circle* dataset, the MSE of the domain indices inferred by GMDI is significantly lower than that inferred by VDI. On *TPT-48* dataset, the domain indices for E (6) → W (42) and N (24) → S (24) regression tasks correspond to longitude and latitude of 48 states. Therefore, we use longitude and latitude as the ground-truth domain indices and calculate the corresponding MSE. In Figure 4, it is evident that the MSE of the domain indices inferred by GMDI is still substantially lower than that of VDI. Although the *CompCars* lacks ground-truth domain indices, the data encoding visualization of VDI and GMDI(Figure 7) show that data encoding generated by GMDI form more distinct clusters compared to VDI. It indirectly indicates the effectiveness of the domain indices inferred by GMDI, demonstrating the considerable impact of modeling the global domain indices as Gaussian Mixture Model. **On all datasets, the domain indices inferred by GMDI outperform those by VDI, owing to the dynamic mixture of domain indices distributions.**

**RQ3: Number of mixture components.** We utilize the stick-breaking representation of the CRP to improve computational efficiency, setting an upper bound $K$ on the number of components in GMM. To study the impact of the number of components on domain adaptation performance, we report classification accuracy on *CompCars*, more complex dataset for different values of K. For other 4 datasets, the best results are achieved with $K = 2$, while for *CompCars*, the best performance

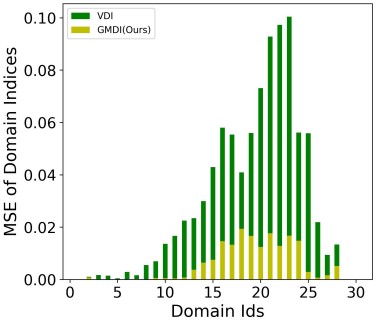
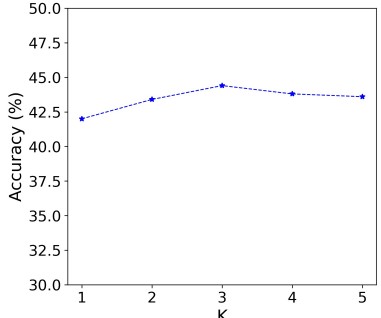

Figure 5: MSE of domain indices on *Circle* dataset.

Figure 6: Accuracy (%) on *CompCars* with different K.

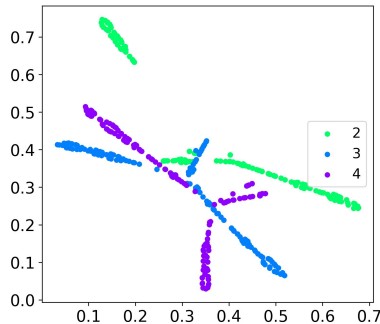
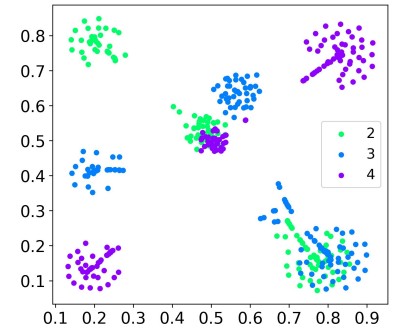

Figure 7: t-SNE visualization of data encoding on *CompCars* dataset. Colors indicating different domains$\{2, 3, 4\}$. **Left**: data encoding generated by VDI. **Right**: data encoding generated by GMDI.

is obtained with $K = 3$. The results in Figure 6 show that accuracy is the lowest when $K = 1$, suggesting that maintaining a single domain index distribution is insufficient for diverse target domains. The choice of $K$ is related to the concentration parameter of CRP and the dataset; the larger the concentration parameter and the more complex the dataset, the larger the value of $K$ should be.

## 7 Conclusion and Limitations

In this work, we propose GMDI, a novel Gaussian Mixture Domain-Indexing algorithm, to address the challenge of inferring domain indices when they are unavailable. Unlike existing methods that assume global domain indices are sampled from a single static Gaussian, GMDI is the first one to utilize a mixture of dynamic Gaussians. The number of mixture components is determined adaptively by the Chinese Restaurant Process, enhancing the flexibility and effectiveness of domain adaptation. Our theoretical analysis confirms that GMDI achieves a more stringent evidence lower bound, closer to the log-likelihood. Extensive experiments validate the effectiveness of GMDI in inferring domain indices and highlight its potential practical applications. Specifically, for classification tasks, GMDI outperforms all approaches, and surpasses the state-of-the-art method, VDI, by up to 3.4%, reaching 99.3%. For regression tasks, GMDI reduces MSE by at least 16% (from 2.496 to 2.087) and by 21% (from 3.160 to 2.493), achieving the lowest errors among all methods. Despite these advantages, GMDI still relies on the availability of domain identities and cannot infer them as latent variables. Future work will focus on developing algorithms capable of inferring domain indices together with domain identities to further enhance the robustness and applicability of our approach.

## Acknowledgments

This research was partly supported by the Fundamental Research Funds for the Central Universities, Sun Yat-sen University (67000-31610047).

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

# A    Finite Stick-Breaking Construction of CRP

The infinite Chinese Restaurant Process (CRP) requires substantial computational overhead. To leverage CRP with lower computational cost, we use the stick-breaking construction to construct it. We set an upper bound on the number of Gaussian mixture components, eliminating the need for a varying number of mixture components. The finite stick-breaking construction of CRP is given as follows:

$$
\begin{aligned}
\beta_k \mid \alpha &\sim \mathrm{Beta}(1, \alpha) \quad \text{for } k\text{=}1, ..., K-1 ~, \\
\pi_k &= \beta_k \prod_{l=1}^{k-1}(1-\beta_l) \quad \text{for } k\text{=}1, ..., K-1 ~, \\
\pi_K &= \prod_{l=1}^{K-1}(1-\beta_l) ~,
\end{aligned}
\tag{21}
$$

where $\boldsymbol{\pi} = \mathrm{stickbreak}(\boldsymbol{\theta})$ is the prior parameters of the $K$-dimensional category variable $v$. Considering the finite stick-breaking construction of CRP mentioned above, we rewrite the generative process of GMDI:

$$
\begin{aligned}
v \mid \boldsymbol{\pi} &\sim \mathrm{Categorical}_K(\boldsymbol{\pi}) ~, \\
\boldsymbol{\theta}^{v=k} &\sim \mathcal{N}(\boldsymbol{\mu}_k, \boldsymbol{\sigma}_k^2) ~, \\
\boldsymbol{u} \mid \boldsymbol{\theta}^{v=k} &\sim p(\boldsymbol{u} \mid \boldsymbol{\theta}^{v=k}) ~, \\
\boldsymbol{x} \mid \boldsymbol{u} &\sim p(\boldsymbol{x} \mid \boldsymbol{u}) ~, \\
\boldsymbol{z} \mid \boldsymbol{x}, \boldsymbol{u}, \boldsymbol{\theta}^v &\sim p(\boldsymbol{z} \mid \boldsymbol{x}, \boldsymbol{u}, \boldsymbol{\theta}^v) ~,
\end{aligned}
\tag{22}
$$

where $\boldsymbol{\mu}_k$ and $\boldsymbol{\sigma}_k^2$ are mean vector and semi-positive covariance matrix of the $k$-th component in GMM, respectively.

# B    Derivation for Variational Posterior

Firstly, we factorize the generative model as follows:

$$
p(\boldsymbol{x}, y, \boldsymbol{u}, \boldsymbol{\theta}, \boldsymbol{z}, v, \boldsymbol{\beta} \mid \alpha) = p(\boldsymbol{\beta} \mid \alpha)p(v \mid \boldsymbol{\beta})p(\boldsymbol{\theta}^v)p(\boldsymbol{u} \mid \boldsymbol{\theta}^v)p(\boldsymbol{x} \mid \boldsymbol{u})p(\boldsymbol{z} \mid \boldsymbol{x}, \boldsymbol{u}, \boldsymbol{\theta}^v)p(y \mid \boldsymbol{z}) ~. \tag{23}
$$

During the inference process, we need to infer the latent variables. Since the target posterior distributions are intractable, we employ the technique of approximate variational inference. In this framework, we design variational distributions for these latent variables, aiming to approximate their true underlying posterior distributions:

$$
q(\boldsymbol{u}, \boldsymbol{\theta}, \boldsymbol{z}, v, \boldsymbol{\beta} \mid \boldsymbol{x}) = q(\boldsymbol{\beta}; \boldsymbol{\gamma})q(v; \boldsymbol{\eta})q(\boldsymbol{u} \mid \boldsymbol{x}; \boldsymbol{\psi}_u)q(\boldsymbol{\theta}^v \mid \boldsymbol{u}; \boldsymbol{\psi}_\theta)q(\boldsymbol{z} \mid \boldsymbol{x}, \boldsymbol{u}, \boldsymbol{\theta}^v; \boldsymbol{\psi}_z) ~. \tag{24}
$$

Thus we calculate the ELBO as follows:

$$
\begin{aligned}
\log p(\boldsymbol{x}, y \mid \alpha) &= \log \int p(\boldsymbol{x}, y, \boldsymbol{u}, \boldsymbol{\theta}, \boldsymbol{z}, v, \boldsymbol{\beta} \mid \alpha)d\boldsymbol{z}d\boldsymbol{u}d\boldsymbol{\theta}dvd\boldsymbol{\beta} \\
&= \log \int q(\boldsymbol{u}, \boldsymbol{\theta}, \boldsymbol{z}, v, \boldsymbol{\beta} \mid \boldsymbol{x}) * \frac{p(\boldsymbol{x}, y, \boldsymbol{u}, \boldsymbol{\theta}, \boldsymbol{z}, v, \boldsymbol{\beta} \mid \alpha)}{q(\boldsymbol{u}, \boldsymbol{\theta}, \boldsymbol{z}, v, \boldsymbol{\beta} \mid \boldsymbol{x})}d\boldsymbol{z}d\boldsymbol{u}d\boldsymbol{\theta}dv \\
&= \log \mathbb{E}_q \left[ \frac{p(\boldsymbol{x}, y, \boldsymbol{u}, \boldsymbol{\theta}, \boldsymbol{z}, v, \boldsymbol{\beta} \mid \alpha)}{q(\boldsymbol{u}, \boldsymbol{\theta}, \boldsymbol{z}, v, \boldsymbol{\beta} \mid \boldsymbol{x})} \right] \\
&\geq \mathbb{E}_q \left[ \log \frac{p(\boldsymbol{x}, y, \boldsymbol{u}, \boldsymbol{\theta}, \boldsymbol{z}, v, \boldsymbol{\beta} \mid \alpha)}{q(\boldsymbol{u}, \boldsymbol{\theta}, \boldsymbol{z}, v, \boldsymbol{\beta} \mid \boldsymbol{x})} \right] \\
&= \mathbb{E}_q \left[ \log \frac{p(\boldsymbol{\beta} \mid \alpha)p(v \mid \boldsymbol{\beta})p(\boldsymbol{u} \mid \boldsymbol{\theta}^v)p(\boldsymbol{x} \mid \boldsymbol{u})p(\boldsymbol{z} \mid \boldsymbol{x}, \boldsymbol{u}, \boldsymbol{\theta}^v)p(y \mid \boldsymbol{z})}{q(\boldsymbol{\beta}; \boldsymbol{\gamma})q(v; \boldsymbol{\eta})q(\boldsymbol{u} \mid \boldsymbol{x}; \boldsymbol{\psi}_u)q(\boldsymbol{\theta}^v \mid \boldsymbol{u}; \boldsymbol{\psi}_\theta)q(\boldsymbol{z} \mid \boldsymbol{x}, \boldsymbol{u}, \boldsymbol{\theta}^v; \boldsymbol{\psi}_z)} \right] \\
&= \mathbb{E}_{q(\boldsymbol{u}, \boldsymbol{\theta}^v, \boldsymbol{z} \mid \boldsymbol{x}; \phi)q(v; \boldsymbol{\eta})}[\log p(y \mid \boldsymbol{z})] + \mathbb{E}_{q(\boldsymbol{u} \mid \boldsymbol{x}; \boldsymbol{\psi}_u)}[\log p(\boldsymbol{x} \mid \boldsymbol{u})] \\
&\quad + \mathbb{E}_{q(v; \boldsymbol{\eta})q(\boldsymbol{\beta}; \boldsymbol{\gamma})q(\boldsymbol{u} \mid \boldsymbol{x}; \boldsymbol{\psi}_u)q(\boldsymbol{\theta}^v \mid \boldsymbol{u}; \boldsymbol{\psi}_\theta)}[\log p(\boldsymbol{u} \mid \boldsymbol{\theta}^v)] \\
&\quad - \mathrm{KL}[q(\boldsymbol{\beta}; \boldsymbol{\gamma})||p(\boldsymbol{\beta})] - \mathbb{E}_{q(\boldsymbol{\beta}; \boldsymbol{\gamma})}[\mathrm{KL}[q(v; \boldsymbol{\eta})||p(v \mid \boldsymbol{\beta})]] \\
&\quad - \mathbb{E}_{q(\boldsymbol{u} \mid \boldsymbol{x}; \boldsymbol{\psi}_u)q(v; \boldsymbol{\eta})}[\mathrm{KL}[q(\boldsymbol{\theta}^v \mid \boldsymbol{u}; \boldsymbol{\psi}_\theta)||p(\boldsymbol{\theta}^v)]] \\
&\quad - \mathbb{E}_{q(v; \boldsymbol{\eta})q(\boldsymbol{u}, \boldsymbol{\theta}^v \mid \boldsymbol{x}; \boldsymbol{\xi})}[\mathrm{KL}[q(\boldsymbol{z} \mid \boldsymbol{x}, \boldsymbol{u}, \boldsymbol{\theta}^v; \boldsymbol{\psi}_z)||p(\boldsymbol{z} \mid \boldsymbol{x}, \boldsymbol{u}, \boldsymbol{\theta}^v)]] \\
&\quad - \mathbb{E}_{q(\boldsymbol{u} \mid \boldsymbol{x}; \boldsymbol{\psi}_u)}[\log(q(\boldsymbol{u} \mid \boldsymbol{x}; \boldsymbol{\psi}_u))] \triangleq \mathcal{L}_{\mathrm{ELBO}} ~,
\end{aligned}
\tag{25}
$$

where the inequality is given by applying Jensen's inequality. Besides, we apply the adversarial loss to ensure the independence between global domain index and data encoding. The adversarial loss with a discriminator D is designed as follows:

$$\mathcal{L}_{\mathrm{D}} = \mathbb{E}_{p(w,\boldsymbol{x})}\mathbb{E}_{q(\boldsymbol{z}|\boldsymbol{x};\boldsymbol{\psi}_z)}[\log \mathrm{D}(w \mid \boldsymbol{z})]. \tag{26}$$

Our final objective is derived as:

$$\mathcal{L}_{\mathrm{GMDI}} = \max \min_{D} \mathcal{L}_{\mathrm{ELBO}} - \lambda * \mathcal{L}_{\mathrm{D}}. \tag{27}$$

By optimizing the objective function, we can calculate all the optimal variational distributions of latent variables as follows:

**Variational distribution of $\boldsymbol{\beta}$.** To obtain the variational posterior of the latent variable $\boldsymbol{\beta}$, We only need to consider the related terms in $\mathcal{L}_{\mathrm{GMDI}}$. Note that the adversarial loss $\mathcal{L}_{\mathrm{D}}$ is independent of the target latent variable, with all terms pertaining to $\boldsymbol{\beta}$ encompassed within the ELBO loss, which can be formulated as follows:

$$F(q(\boldsymbol{\beta}_k; \boldsymbol{\gamma}_k)) = \mathbb{E}_{q(v;\boldsymbol{\eta})}\mathbb{E}_{q(\boldsymbol{\beta}_k;\boldsymbol{\gamma}_k)}[\log p(\boldsymbol{\beta}_k) + \log p(v)] - \mathbb{E}_{q(\boldsymbol{\beta}_k;\boldsymbol{\gamma}_k)}[\log p(\boldsymbol{\beta}_k)]. \tag{28}$$

Subsequently, by differentiating the function and setting the derivative equal to zero:

$$\frac{\partial}{\partial q(\boldsymbol{\beta}_k;\boldsymbol{\gamma}_k)} F(q(\boldsymbol{\beta}_k;\boldsymbol{\gamma}_k))$$

$$= \int q(v;\boldsymbol{\eta})q(\boldsymbol{\beta}_k;\boldsymbol{\gamma}_{\backslash k}) \left[ \frac{\partial}{\partial q(\boldsymbol{\beta}_k;\boldsymbol{\gamma}_k)} \int q(\boldsymbol{\beta}_k;\boldsymbol{\gamma}_k)[\log p(\boldsymbol{\beta}_k) + \log p(v \mid \boldsymbol{\beta}) - \log q(\boldsymbol{\beta}_k;\boldsymbol{\gamma}_k)]d\boldsymbol{\beta}_k \right] d\boldsymbol{\beta}_{\backslash k}dv$$

$$= \int q(v;\boldsymbol{\eta})q(\boldsymbol{\beta}_k;\boldsymbol{\gamma}_{\backslash k})[\log p(\boldsymbol{\beta}_k) + \log p(v \mid \boldsymbol{\beta}) - \log q(\boldsymbol{\beta}_k;\boldsymbol{\gamma}_k)]d\boldsymbol{\beta}_{\backslash k}dv,$$
$$\tag{29}$$

where $q(\boldsymbol{\beta}_k;\boldsymbol{\gamma}_{\backslash k})$ is the variational posterior of $\boldsymbol{\beta}$ without $\boldsymbol{\beta}_k$, we derive the final variational distribution of $\boldsymbol{\beta}_k$ as follows:

$$\log q(\boldsymbol{\beta}_k;\boldsymbol{\gamma}_k) \propto \mathbb{E}_{q(v;\boldsymbol{\eta})}\mathbb{E}_{q(\boldsymbol{\beta}_{\backslash k};\boldsymbol{\gamma}_{\backslash k})}[\log p(\boldsymbol{\beta}_k) + \log p(v \mid \boldsymbol{\beta})]$$

$$= \log p(\boldsymbol{\beta}_k) + \mathbb{E}_{q(v;\boldsymbol{\eta})}\left[\mathbb{E}_{q(\boldsymbol{\beta}_{\backslash k};\boldsymbol{\gamma}_{\backslash k})}\left[ \sum_{k=1}^{K-1}\left( \boldsymbol{v}_k \log \boldsymbol{\beta}_k + \sum_{l=1}^{k-1} \boldsymbol{v}_k \log(1-\boldsymbol{\beta}_l) \right) \right.\right.$$

$$\left.\left. + \sum_{j=1}^{K} v_K \log(1-\boldsymbol{\beta}_j) \right]\right]$$

$$\propto \log p(\boldsymbol{\beta}_k) + \mathbb{E}_{q(v;\boldsymbol{\eta})}\left[ v_k \log \boldsymbol{\beta}_k + \sum_{j=k+1}^{K-1} v_j \log(1-\boldsymbol{\beta}_k) + v_K \log(1-\boldsymbol{\beta}_k) \right]$$

$$= \log p(\boldsymbol{\beta}_k) + \boldsymbol{\eta}_k \log \boldsymbol{\beta}_k + \sum_{j=k+1}^{K} \boldsymbol{\eta}_j \log(1-\boldsymbol{\beta}_k). \tag{30}$$

Since we assume that $p(\boldsymbol{\beta}_k) \sim Beta(\cdot; 1, \alpha)$, we have:

$$\log p(\boldsymbol{\beta}_k) \propto (\alpha - 1)\log(1 - \boldsymbol{\beta}_k). \tag{31}$$

Thus we can derive the optimal variational posterior of $\boldsymbol{\beta}_k$ as:

$$\log q(\boldsymbol{\beta}_k;\boldsymbol{\gamma}_k) \propto \boldsymbol{\eta}_k \log \boldsymbol{\beta}_k + (\alpha - 1 + \sum_{j=k+1}^{K} \boldsymbol{\eta}_j)\log(1 - \boldsymbol{\beta}_k), \tag{32}$$

which is also a Beta distribution $\mathrm{Beta}(\boldsymbol{\beta}_k; \boldsymbol{\gamma}_{k,1}, \boldsymbol{\gamma}_{k,2})$ with parameters:

$$\boldsymbol{\gamma}_{k,1} = 1 + \boldsymbol{\eta}_k, \quad \boldsymbol{\gamma}_{k,2} = \alpha + \sum_{j=k+1}^{K} \boldsymbol{\eta}_j. \tag{33}$$

**Variational distribution of $v$.** Since $v$ is a category variable, we assume its variational posterior to be a categorical distribution parameterized by $\boldsymbol{\eta}$ as:

$$q(v;\boldsymbol{\eta}) = \mathrm{Categorical}_K(v;\boldsymbol{\eta}). \tag{34}$$

Similarly, we only consider the terms related to $v$ in $\mathcal{L}_{\mathrm{GMDI}}$ to derive the optimal varational posterior, which is calculated as:

$$
\begin{aligned}
F(q(v;\boldsymbol{\eta})) = {} & -\mathbb{E}_{q(\boldsymbol{\beta};\boldsymbol{\gamma})}[\mathrm{KL}[q(v;\boldsymbol{\eta})||p(v|\boldsymbol{\beta})]] - \mathbb{E}_{q(\boldsymbol{u}|\boldsymbol{x};\boldsymbol{\psi}_u)q(v;\boldsymbol{\eta})}[\mathrm{KL}[q(\boldsymbol{\theta}^v|\boldsymbol{u};\boldsymbol{\psi}_\theta)||p(\boldsymbol{\theta}^v)]] \\
& + \mathbb{E}_{q(v;\boldsymbol{\eta})q(\boldsymbol{u}|\boldsymbol{x};\boldsymbol{\psi}_u)q(\boldsymbol{\theta}^v|\boldsymbol{u};\boldsymbol{\psi}_\theta)}[\log p(\boldsymbol{u}|\boldsymbol{\theta}^v)] \\
& - \mathbb{E}_{q(\boldsymbol{u},\boldsymbol{\theta},v|\boldsymbol{x};\boldsymbol{\xi})}[\mathrm{KL}[q(\boldsymbol{z}|\boldsymbol{x},\boldsymbol{u},\boldsymbol{\beta},v;\boldsymbol{\psi}_z)||p(\boldsymbol{z}|\boldsymbol{x},\boldsymbol{u},\boldsymbol{\theta},v)]] \\
= {} & \sum_{k=1}^{K} \left\{ \boldsymbol{\eta}_k \mathbb{E}_{q(\boldsymbol{\beta};\boldsymbol{\gamma})}[\log p(v|\boldsymbol{\beta})] - \boldsymbol{\eta}_k \log \boldsymbol{\eta}_k - \boldsymbol{\eta}_k \mathbb{E}_{q(\boldsymbol{u}|\boldsymbol{x};\boldsymbol{\psi}_u)}[\mathrm{KL}[q(\boldsymbol{\theta}^v|\boldsymbol{u};\boldsymbol{\psi}_\theta)||p(\boldsymbol{\theta}^v)]] \right. \\
& + \boldsymbol{\eta}_k \mathbb{E}_{q(\boldsymbol{u}|\boldsymbol{x};\boldsymbol{\psi}_u)q(\boldsymbol{\theta}^v|\boldsymbol{u};\boldsymbol{\psi}_\theta)}[\log p(\boldsymbol{u}|\boldsymbol{\theta}^v)] \\
& \left. - \boldsymbol{\eta}_k \mathbb{E}_{q(\boldsymbol{u},\boldsymbol{\theta}^v|\boldsymbol{x};\boldsymbol{\xi})}[\mathrm{KL}[q(\boldsymbol{z}|\boldsymbol{x},\boldsymbol{u},\boldsymbol{\theta}^v;\boldsymbol{\psi}_\theta)||p(\boldsymbol{z}|\boldsymbol{x},\boldsymbol{u},\boldsymbol{\theta}^v)]] \right\}.
\end{aligned}
\tag{35}
$$

By taking the derivative function of $F(q(v;\boldsymbol{\eta}))$ with respective to zero:

$$
\begin{aligned}
\frac{\partial}{\partial \boldsymbol{\eta}_k} F(q(v;\boldsymbol{\eta})) = {} & \mathbb{E}_{q(\boldsymbol{\beta};\boldsymbol{\gamma})}[\log p(v|\boldsymbol{\beta})] - \log \boldsymbol{\eta}_k - 1 - \mathbb{E}_{q(\boldsymbol{u}|\boldsymbol{x};\boldsymbol{\psi}_u)}[\mathrm{KL}[q(\boldsymbol{\theta}^v|\boldsymbol{u};\boldsymbol{\psi}_\theta)||p(\boldsymbol{\theta}^v)]] \\
& + \mathbb{E}_{q(\boldsymbol{u}|\boldsymbol{x};\boldsymbol{\psi}_u)q(\boldsymbol{\theta}^v|\boldsymbol{u};\boldsymbol{\psi}_\theta)}[\log p(\boldsymbol{u}|\boldsymbol{\theta}^v)] \\
& - \mathbb{E}_{q(\boldsymbol{u},\boldsymbol{\theta}^v|\boldsymbol{x};\boldsymbol{\xi})}[\mathrm{KL}[q(\boldsymbol{z}|\boldsymbol{x},\boldsymbol{u},\boldsymbol{\theta}^v;\boldsymbol{\psi}_z)||p(\boldsymbol{z}|\boldsymbol{x},\boldsymbol{u},\boldsymbol{\theta}^v)]],
\end{aligned}
\tag{36}
$$

we can finally have:

$$
\begin{aligned}
\log \boldsymbol{\eta}_k \propto {} & \mathbb{E}_{q(\boldsymbol{\beta};\boldsymbol{\gamma})}[\boldsymbol{\pi}] + \mathbb{E}_{q(\boldsymbol{u}|\boldsymbol{x};\boldsymbol{\psi}_u)q(\boldsymbol{\theta}^v|\boldsymbol{u};\boldsymbol{\psi}_\theta)}[\log p(\boldsymbol{u}|\boldsymbol{\theta}^v)] \\
& - \mathbb{E}_{q(\boldsymbol{u}|\boldsymbol{x};\boldsymbol{\psi}_u)}[\mathrm{KL}[q(\boldsymbol{\theta}^v|\boldsymbol{u};\boldsymbol{\psi}_\theta)||p(\boldsymbol{\theta}^v)]] \\
& - \mathbb{E}_{q(\boldsymbol{u},\boldsymbol{\theta}^v|\boldsymbol{x};\boldsymbol{\xi})}[\mathrm{KL}[q(\boldsymbol{z}|\boldsymbol{u},\boldsymbol{\theta},\boldsymbol{x};\boldsymbol{\psi}_z)||p(\boldsymbol{z}|\boldsymbol{u},\boldsymbol{\theta},\boldsymbol{x})]],
\end{aligned}
\tag{37}
$$

where $\sum_{k=1}^{K} \boldsymbol{\eta}_k = 1$ and $q(\boldsymbol{\beta};\boldsymbol{\gamma}) = \prod_{k=1}^{K-1} q(\boldsymbol{\beta}_k;\boldsymbol{\gamma}_k)$.

**Variational distribution of $\boldsymbol{\theta}, \boldsymbol{u}$ and $\boldsymbol{z}$.** The distribution of $\boldsymbol{\theta}$ is assumed to include a mixture of components. The distribution of the latent variable $\boldsymbol{\theta}$ is assumed to be a mixture of a series of distributions, while the latent variables $\boldsymbol{u}$ and $\boldsymbol{z}$ can be regarded as following conditional Gaussian distributions. Since it is highly intractable to precisely compute the posterior distributions of these latent variables, we employ variational Gaussian distributions to approximate the posterior distribution for each component of $\boldsymbol{\theta}$, as well as the conditional posterior distributions for $\boldsymbol{u}$ and $\boldsymbol{z}$:

$$
q(\boldsymbol{\theta}^{v=k}|\boldsymbol{u};\boldsymbol{\psi}_\theta) = \mathcal{N}(\boldsymbol{\mu}_\theta^k, \boldsymbol{\Lambda}_\theta^k),
\tag{38}
$$

$$
q(\boldsymbol{u}|\boldsymbol{x};\boldsymbol{\psi}_u) = \mathcal{N}(\boldsymbol{\mu}_u, \boldsymbol{\Lambda}_u),
\tag{39}
$$

$$
q(\boldsymbol{z}|\boldsymbol{x},\boldsymbol{u},\boldsymbol{\theta}^v;\boldsymbol{\psi}_z) = \mathcal{N}(\boldsymbol{\mu}_z, \boldsymbol{\Lambda}_z),
\tag{40}
$$

where $\boldsymbol{\mu}$ and $\boldsymbol{\Lambda}$ are mean vector and semi-positive covariance matrix of Gaussian distribution. All these variational distributions can be updated by maximizing $\mathcal{L}_{\mathrm{GMDI}}$ with back propagation.

## C Theory Analysis

The proof process of the lemmas and theorems are partially based on VDI [41].

**Lemma C.1** *The ELBO of $p(\boldsymbol{x}, y)$ is bounded by the following formula with the Mutual Information, the Entropy and the* KL*-divergence:*

$$
\begin{aligned}
\mathbb{E}_{p(\boldsymbol{x},y)}[\mathcal{L}_{\mathrm{ELBO}}(p(\boldsymbol{x},y))] \leq {} & I(y;\boldsymbol{z}) + I(\boldsymbol{x};\boldsymbol{u},\boldsymbol{\theta},\boldsymbol{z},v) - (H(\boldsymbol{x}) + H(y)) \\
& - \mathbb{E}_{q(\boldsymbol{x},\boldsymbol{u},\boldsymbol{\theta},\boldsymbol{z},v)}[\mathrm{KL}[q(\boldsymbol{x}|\boldsymbol{u},\boldsymbol{\theta},v,\boldsymbol{z})||p(\boldsymbol{x}|\boldsymbol{u},\boldsymbol{\theta},v,\boldsymbol{z})]] \\
& - \mathrm{KL}[q(\boldsymbol{u},\boldsymbol{\theta},v,\boldsymbol{z}|\boldsymbol{x})||p(\boldsymbol{u},\boldsymbol{\theta},\boldsymbol{z},v)].
\end{aligned}
$$

The main difference between and Lemma C.1 in GMDI and Lemma B.1 in VDI [41] is the last two KL terms and the inclusion of $v$.

*Proof.* In order to give an upper bound of $p(\boldsymbol{x}, y)$, we first calculate the ELBO as follows:

$$\log p(\boldsymbol{x}, y) = \log \int p(\boldsymbol{x}, \boldsymbol{u}, \boldsymbol{\theta}, v, \boldsymbol{\beta}, \boldsymbol{z}, y) dz d\boldsymbol{u} d\boldsymbol{\theta}^v dv d\boldsymbol{\beta}$$

$$= \log \int \frac{p(\boldsymbol{x}, \boldsymbol{u}, \boldsymbol{\theta}, v, \boldsymbol{\beta}, \boldsymbol{z}, y) * q(\boldsymbol{u}, \boldsymbol{\theta}, v, \boldsymbol{\beta}, \boldsymbol{z}|\boldsymbol{x})}{q(\boldsymbol{u}, \boldsymbol{\theta}, v, \boldsymbol{\beta}, \boldsymbol{z}|\boldsymbol{x})} dz d\boldsymbol{u} d\boldsymbol{\theta}^v dv d\boldsymbol{\beta}$$

$$= \log \mathbb{E}_q \left[ \frac{p(\boldsymbol{x}, \boldsymbol{u}, \boldsymbol{\theta}, v, \boldsymbol{\beta}, \boldsymbol{z}, y)}{q(\boldsymbol{u}, \boldsymbol{\theta}, v, \boldsymbol{\beta}, \boldsymbol{z}|\boldsymbol{x})} \right]$$

$$\geq \mathbb{E}_q \left[ \log \frac{p(\boldsymbol{x}, \boldsymbol{u}, \boldsymbol{\theta}, v, \boldsymbol{\beta}, \boldsymbol{z}, y)}{q(\boldsymbol{u}, \boldsymbol{\theta}, v, \boldsymbol{\beta}, \boldsymbol{z}|\boldsymbol{x})} \right]$$

$$= \mathbb{E}_q \left[ \log \frac{p(y|\boldsymbol{z}) p(\boldsymbol{x}|\boldsymbol{u}, \boldsymbol{\theta}, v, \boldsymbol{\beta}, \boldsymbol{z}) p(\boldsymbol{u}, \boldsymbol{\theta}, v, \boldsymbol{\beta}, \boldsymbol{z})}{q(\boldsymbol{u}, \boldsymbol{\theta}, v, \boldsymbol{\beta}, \boldsymbol{z}|\boldsymbol{x})} \right]$$

$$= \mathbb{E}_q[\log p(y|\boldsymbol{z})] + \mathbb{E}_q[\log p(\boldsymbol{x}|\boldsymbol{u}, \boldsymbol{\theta}, v, \boldsymbol{\beta}, \boldsymbol{z})] - \mathrm{KL}[p(\boldsymbol{u}, \boldsymbol{\theta}, v, \boldsymbol{\beta}, \boldsymbol{z}) || q(\boldsymbol{u}, \boldsymbol{\theta}, v, \boldsymbol{\beta}, \boldsymbol{z}|\boldsymbol{x})].$$

Accordingly, we have the following ELBO objective:

$$\mathcal{L}_{\mathrm{ELBO}}(p(\boldsymbol{x}, y)) = \mathbb{E}_q[\log p(y|\boldsymbol{z})] + \mathbb{E}_q[\log p(\boldsymbol{x}|\boldsymbol{u}, \boldsymbol{\theta}, v, \boldsymbol{\beta}, \boldsymbol{z})] - \mathrm{KL}[p(\boldsymbol{u}, \boldsymbol{\theta}, v, \boldsymbol{\beta}, \boldsymbol{z}) || q(\boldsymbol{u}, \boldsymbol{\theta}, v, \boldsymbol{\beta}, \boldsymbol{z}|\boldsymbol{x})]. \tag{41}$$

Here we aim at giving a formula only including **Mutual Information**, **Entropy** and **KL-divergence** to bound the objective. To achieve this, we first calculate the upper bound of $\mathbb{E}_q \log p(y|\boldsymbol{z})$, with denoting $r(\boldsymbol{x}, y, \boldsymbol{z}) = p(\boldsymbol{x}, y) q(\boldsymbol{z}|\boldsymbol{x})$:

$$\mathbb{E}_{p(\boldsymbol{x}, y)} \mathbb{E}_q[\log p(y|\boldsymbol{z})] = \mathbb{E}_{p(\boldsymbol{x}, y) q(\boldsymbol{z}|\boldsymbol{x})}[\log p(y|\boldsymbol{z})] \tag{42}$$

$$= \mathbb{E}_{r(\boldsymbol{x}, y, \boldsymbol{z})}[\log p(y|\boldsymbol{z})] \tag{43}$$

$$= \mathbb{E}_{r(y, \boldsymbol{z})}[\log p(y|\boldsymbol{z})] \tag{44}$$

$$\leq \mathbb{E}_{r(y, \boldsymbol{z})}[\log r(y|\boldsymbol{z})] \tag{45}$$

$$= \mathbb{E}_{r(y, \boldsymbol{z})}[\log \frac{r(y|\boldsymbol{z})}{p(y)}] + \mathbb{E}_{r(y, \boldsymbol{z})}[\log p(y)] \tag{46}$$

$$= I(y|\boldsymbol{z}) - H(y). \tag{47}$$

When $p(y|\boldsymbol{z}) = r(y|\boldsymbol{z})$, the maximum of $\mathbb{E}_q[\log p(y|\boldsymbol{z})]$ is achieved and then we have $\max \mathbb{E}_q[\log p(y|\boldsymbol{z})] = I(y; \boldsymbol{z}) - H(y)$.

Secondly, we need to give an upper bound of $\mathbb{E}_q[\log p(\boldsymbol{x}|\boldsymbol{u}, \boldsymbol{\theta}, v, \boldsymbol{\beta}, \boldsymbol{z})]$. For convenient, we denote the joint distribution $s(\boldsymbol{x}, \boldsymbol{u}, \boldsymbol{\theta}, v, \boldsymbol{\beta}, \boldsymbol{z}) = p(\boldsymbol{x}) q(\boldsymbol{u}, \boldsymbol{\theta}, v, \boldsymbol{\beta}, \boldsymbol{z})$, then we can calculate as follows:

$$\mathbb{E}_{p(\boldsymbol{x}, y)} \mathbb{E}_q[\log p(\boldsymbol{x}|\boldsymbol{u}, \boldsymbol{\theta}, v, \boldsymbol{\beta}, \boldsymbol{z})] = \mathbb{E}_{p(\boldsymbol{x})} \mathbb{E}_q[\log p(\boldsymbol{x}|\boldsymbol{u}, \boldsymbol{\theta}, v, \boldsymbol{\beta}, \boldsymbol{z})]$$

$$= \mathbb{E}_{p(\boldsymbol{x})} \mathbb{E}_q[\log \frac{q(\boldsymbol{x}|\boldsymbol{u}, \boldsymbol{\theta}, v, \boldsymbol{\beta}, \boldsymbol{z})}{p(\boldsymbol{x})} \frac{p(\boldsymbol{x}) p(\boldsymbol{x}|\boldsymbol{u}, \boldsymbol{\theta}, v, \boldsymbol{\beta}, \boldsymbol{z})}{q(\boldsymbol{x}|\boldsymbol{u}, \boldsymbol{\theta}, v, \boldsymbol{\beta}, \boldsymbol{z})}]$$

$$= \mathbb{E}_{p(\boldsymbol{x})} \mathbb{E}_q[\log \frac{q(\boldsymbol{x}|\boldsymbol{u}, \boldsymbol{\theta}, v, \boldsymbol{\beta}, \boldsymbol{z})}{p(\boldsymbol{x})}] + \mathbb{E}_{p(\boldsymbol{x})} \mathbb{E}_q[\log p(\boldsymbol{x})] + \mathbb{E}_{p(\boldsymbol{x})} \mathbb{E}_q[\log \frac{p(\boldsymbol{x}|\boldsymbol{u}, \boldsymbol{\theta}, v, \boldsymbol{\beta}, \boldsymbol{z})}{q(\boldsymbol{x}|\boldsymbol{u}, \boldsymbol{\theta}, v, \boldsymbol{\beta}, \boldsymbol{z})}]$$

$$= I_s(\boldsymbol{x}; \boldsymbol{u}, \boldsymbol{\theta}, v, \boldsymbol{\beta}, \boldsymbol{z}) + H(\boldsymbol{x}) - \mathrm{KL}[q(\boldsymbol{x}|\boldsymbol{u}, \boldsymbol{\theta}, v, \boldsymbol{\beta}, \boldsymbol{z}) || p(\boldsymbol{x}|\boldsymbol{u}, \boldsymbol{\theta}, v, \boldsymbol{\beta}, \boldsymbol{z})].$$

Finally, we apply these two bounds on the Eq. 41 and then we can proove the Lemma C.1:

$$\mathbb{E}_{p(\boldsymbol{x}, y)}[\mathcal{L}_{\mathrm{ELBO}}(p(\boldsymbol{x}, y))] \leq I(y; \boldsymbol{z}) + I(\boldsymbol{x}; \boldsymbol{u}, \boldsymbol{\theta}, \boldsymbol{z}, v) - (H(\boldsymbol{x}) + H(y))$$
$$- \mathbb{E}_{q(\boldsymbol{x}, \boldsymbol{u}, \boldsymbol{\theta}, \boldsymbol{z}, v)}[\mathrm{KL}[q(\boldsymbol{x}|\boldsymbol{u}, \boldsymbol{\theta}, v, \boldsymbol{z}) || p(\boldsymbol{x}|\boldsymbol{u}, \boldsymbol{\theta}, v, \boldsymbol{z})]]$$
$$- \mathrm{KL}[q(\boldsymbol{u}, \boldsymbol{\theta}, v, \boldsymbol{z}|\boldsymbol{x}) || p(\boldsymbol{u}, \boldsymbol{\theta}, \boldsymbol{z}, v)].$$

**Lemma C.2** *(Information Decomposition of the Adversarial Loss [41]) We can decompose the global maximum of adversial loss as follows:*

$$\max_D \mathbb{E}_{p(w, \boldsymbol{x})} \mathbb{E}_{q(\boldsymbol{z}|\boldsymbol{x})}[\log \mathrm{D}(w|\boldsymbol{z})] = I(\boldsymbol{z}; \boldsymbol{\theta}, v) + I(\boldsymbol{z}, w|\boldsymbol{\theta}, v) - H(w).$$

*The global minimum of the function is achieved if and only if $I(\boldsymbol{z}; \boldsymbol{\theta}) = 0$ and $I(\boldsymbol{z}, w|\boldsymbol{\theta}) = 0$*

*Proof.* With denoting $t(w, \boldsymbol{x}, \boldsymbol{\theta}, \boldsymbol{z}) = p(w, \boldsymbol{x}) q(\boldsymbol{z}|\boldsymbol{x}) q(\boldsymbol{\theta}, v|w)$, we have:

$$\mathbb{E}_{p(w, \boldsymbol{x})} \mathbb{E}_{q(\boldsymbol{z}|\boldsymbol{x})}[\log \mathrm{D}(w|\boldsymbol{z})] = \mathbb{E}_{t(w, \boldsymbol{z})}[\log \mathrm{D}(w|\boldsymbol{z})] \leq \mathbb{E}_{t(w, \boldsymbol{z})}[\log t(w|\boldsymbol{z})], \tag{48}$$

and $\mathbb{E}_{p(w,\boldsymbol{x})}\mathbb{E}_{q(\boldsymbol{z}|\boldsymbol{x})}[\log \mathrm{D}(w|\boldsymbol{z})]$ achieves the maximum when $t(w,\boldsymbol{x},\boldsymbol{\theta},\boldsymbol{z}) = \mathrm{D}(w|\boldsymbol{z})$. To futher analyze the joint distribution $t(w,\boldsymbol{\theta},\boldsymbol{z}) := q(\boldsymbol{z}|\boldsymbol{\theta},v,w)q(\boldsymbol{\theta},v|w)p(w)$, we assume that there is a function $v(w)$ mapping $w$ to a group of domain-related weights $v$, then we can have:

$$p(\boldsymbol{z}|\boldsymbol{\theta},v,w) = p(\boldsymbol{z}|\boldsymbol{\theta},v(w),w) = p(\boldsymbol{z}|\boldsymbol{\theta}^{v(w)}) = p(\boldsymbol{z}|w). \tag{49}$$

Accordingly, we can factorize the joint distribution $t(w,\boldsymbol{\theta},v,\boldsymbol{z}) = q(\boldsymbol{z}|w)q(\boldsymbol{\theta}^v|w)p(w)$. Therefore, we can factorize $I(\boldsymbol{z};\boldsymbol{\theta},v,w)$ into two different styles with the chain rule for mutual information:

$$I(\boldsymbol{z};\boldsymbol{\theta},v) + I(\boldsymbol{z},w|\boldsymbol{\theta},v) = I(\boldsymbol{z};\boldsymbol{\theta},v,w) = I(\boldsymbol{z};w) + I_q(\boldsymbol{z},\boldsymbol{\theta},v|w), \tag{50}$$

where $I_q(\boldsymbol{z},\boldsymbol{\theta},v|w) = 0$ due to the chain rule. That means:

$$I(z;w) = I(\boldsymbol{z};\boldsymbol{\theta},v) + I(\boldsymbol{z},w|\boldsymbol{\theta},v). \tag{51}$$

And we also have:

$$\mathbb{E}_{t(w,\boldsymbol{z})}[\log t(w|\boldsymbol{z})] = \mathbb{E}_{t(w,\boldsymbol{z})}[\log \frac{t(w|\boldsymbol{z})}{q(w)}] + \mathbb{E}_{t(w,\boldsymbol{z})}[\log q(w)] \tag{52}$$

$$= I(w;\boldsymbol{z}) - H(w). \tag{53}$$

Thus we have:

$$\max_D \mathbb{E}_{p(w,\boldsymbol{x})}\mathbb{E}_{q(\boldsymbol{z}|\boldsymbol{x})}[\log \mathrm{D}(w|\boldsymbol{z})] = I(w;\boldsymbol{z}) - H(w) = I(\boldsymbol{z};\boldsymbol{\theta},v) + I(\boldsymbol{z},w|\boldsymbol{\theta},v) - H(w). \tag{54}$$

Accordingly, $\min\max_D \mathbb{E}_{p(w,\boldsymbol{x})}\mathbb{E}_{q(\boldsymbol{z}|\boldsymbol{x})}[\log \mathrm{D}(w|\boldsymbol{z})] = 0$ if and only if $I(\boldsymbol{z};\boldsymbol{\theta},v) = I(\boldsymbol{z},w|\boldsymbol{\theta},v) = 0$ due to the fact that $I(\cdot) \geq 0$.

**Theorem C.1** *The upper bound of the objective function can be decomposed as follows:*

$$\mathcal{L}_{\mathrm{GMDI}} \leq I(y;\boldsymbol{z}) + I(\boldsymbol{x};\boldsymbol{u},\boldsymbol{\theta},\boldsymbol{z},v) - I(\boldsymbol{z};\boldsymbol{\theta}) - I(\boldsymbol{z};w|\boldsymbol{\theta}) - (H(\boldsymbol{x}) + H(y) - H(w)).$$

The main difference between Theorem C.1 in GMDI and Theorem B.1 in VDI [41] is the inclusion of $v$.

*Proof.* To proove the theorem, we apply the Lemma C.1 and C.2 to directly get the final upper bound:

$$\mathcal{L}_{\mathrm{GMDI}} = \mathbb{E}_{p(\boldsymbol{x},y)}[\mathcal{L}_{\mathrm{ELBO}}(p(\boldsymbol{x},y))] - \max_D \mathbb{E}_{p(w,\boldsymbol{x})}\mathbb{E}_{q(\boldsymbol{z}|\boldsymbol{x})}[\log \mathrm{D}(w|\boldsymbol{z})]$$

$$\leq I(y;\boldsymbol{z}) + I(\boldsymbol{x};\boldsymbol{u},\boldsymbol{\theta},\boldsymbol{z},v) - I(\boldsymbol{z};\boldsymbol{\theta}) - I(\boldsymbol{z};w|\boldsymbol{\theta}) - (H(\boldsymbol{x}) + H(y) - H(w))$$
$$- \mathbb{E}_{q(\boldsymbol{x},\boldsymbol{u},\boldsymbol{\theta},v,\boldsymbol{z})}[\mathrm{KL}[q(\boldsymbol{x}|\boldsymbol{u},\boldsymbol{\theta},v,\boldsymbol{z})||p(\boldsymbol{x}|\boldsymbol{u},\boldsymbol{\theta},v,\boldsymbol{z})]] - \mathrm{KL}[q(\boldsymbol{u},\boldsymbol{\theta},v,\boldsymbol{z}|\boldsymbol{x})||p(\boldsymbol{u},\boldsymbol{\theta},\boldsymbol{z},v)]$$
$$\leq I(y;\boldsymbol{z}) + I(\boldsymbol{x};\boldsymbol{u},\boldsymbol{\theta},\boldsymbol{z},v) - I(\boldsymbol{z};\boldsymbol{\theta}) - I(\boldsymbol{z};w|\boldsymbol{\theta}) - (H(\boldsymbol{x}) + H(y) - H(w)),$$

where the second equality holds when all the terms of KL-divergence are equal to zero.

**Theorem C.2** *The global optimum is achieved if and only if: (1)$I(\boldsymbol{z};\boldsymbol{\theta}) = I(\boldsymbol{z};w|\boldsymbol{\theta}) = 0$, (2)$I(y;\boldsymbol{z})$ and $I(\boldsymbol{x};\boldsymbol{u},\boldsymbol{\theta},\boldsymbol{z},v)$ are maximized,(3)$\mathrm{KL}[q(\boldsymbol{u},\boldsymbol{\theta},v,\boldsymbol{z}|\boldsymbol{x})||p(\boldsymbol{u},\boldsymbol{\theta},v,\boldsymbol{z})] = 0$ and $\mathrm{KL}[q(\boldsymbol{x}|\boldsymbol{u},\boldsymbol{\theta},v,\boldsymbol{z})||p(\boldsymbol{x}|\boldsymbol{u},\boldsymbol{\theta},v,\boldsymbol{z})] = 0$.*

The main difference between Theorem C.2 in GMDI and Theorem B.2 in VDI [41] is that $I(\boldsymbol{x};\boldsymbol{u},\boldsymbol{\theta},\boldsymbol{z},v)$, which includes $v$, needs to be maximized, and the two KL divergences should equal zero.

*Proof.* The theorem can be prooved by observing the conditions from Lemma C.1, C.2 and Theorem C.1.

**Theorem C.3** *Assuming the ELBO and objective of VDI are $\mathcal{L}_{\mathrm{VDI-ELBO}}$ and $\mathcal{L}_{\mathrm{VDI}}$ respectively, where domain indices are sampled from a simple Gaussian prior, we can prove that our objective achieves a more stringent evidence lower bound which is closer to the log-likelihood, and also a tighter upper bound of the objective: $\mathcal{L}_{\mathrm{VDI-ELBO}} \leq \mathcal{L}_{\mathrm{ELBO}} \leq \log p(\boldsymbol{x},y)$ and $\mathcal{L}_{\mathrm{VDI}} \leq \mathcal{L}_{\mathrm{GMDI}}$.*

*Proof.* To compare our objective loss with the VDI's, we first list the ELBO loss of VDI here and provide an upper bound:

$$\mathcal{L}_{\mathrm{VDI-ELBO}}(p(\boldsymbol{x},y)) = \mathbb{E}_q[\log p(y|\boldsymbol{z})] + \mathbb{E}_q[\log p(\boldsymbol{x}|\boldsymbol{u},\hat{\boldsymbol{\theta}},\boldsymbol{z})] - \mathrm{KL}[p(\boldsymbol{u},\hat{\boldsymbol{\theta}},\boldsymbol{z})||q(\boldsymbol{u},\hat{\boldsymbol{\theta}},\boldsymbol{z}|\boldsymbol{x})]$$

$$\leq I(y;\boldsymbol{z}) + I(\boldsymbol{x};\boldsymbol{u},\hat{\boldsymbol{\theta}},\boldsymbol{z},v) - (H(\boldsymbol{x}) + H(y))$$
$$- \mathbb{E}_{q(\boldsymbol{x},\boldsymbol{u},\hat{\boldsymbol{\theta}},\boldsymbol{z})}[\mathrm{KL}[q(\boldsymbol{x}|\boldsymbol{u},\hat{\boldsymbol{\theta}},\boldsymbol{z})||p(\boldsymbol{x}|\boldsymbol{u},\hat{\boldsymbol{\theta}},\boldsymbol{z})]]$$
$$- \mathrm{KL}[q(\boldsymbol{u},\hat{\boldsymbol{\theta}},\boldsymbol{z}|\boldsymbol{x})||p(\boldsymbol{u},\hat{\boldsymbol{\theta}},\boldsymbol{z})].$$

We can observe that the most significant difference is the prior distribution, which mainly affects the term of KL-divergence. To further analyze, it is obvious that when $\boldsymbol{\theta}^{v=k} = \hat{\boldsymbol{\theta}}$, VDI is a special case of our proposed method GMDI. Hence we have:

$$max\mathcal{L}_{\text{VDI}-\text{ELBO}}(p(\boldsymbol{x}, y)) \leq \max \mathcal{L}_{\text{GMDI}-\text{ELBO}}(p(\boldsymbol{x}, y)). \tag{55}$$

Further more, we can notice that the adversarial loss is independent of the prior distribution of global indices and we can have:

$$\begin{aligned}
\mathcal{L}_{\text{VDI}} &= \max \min_D \mathcal{L}_{\text{VDI}-\text{ELBO}}(p(\boldsymbol{x}, y)) - \lambda * \mathcal{L}_{\text{D}} \\
&\leq \max \min_D \mathcal{L}_{\text{GMDI}-\text{ELBO}}(p(\boldsymbol{x}, y)) - \lambda * \mathcal{L}_{\text{D}} \\
&= \mathcal{L}_{\text{GMDI}}.
\end{aligned} \tag{56}$$

## D  Visualization of Inferred Domain Indices for Circle

Figure 8 shows the inferred domain indices for *Circle* dataset. GMDI's inferred indices have a correlation of 0.99 with true indices, even though *GMDI does not have access to true indices during training*.

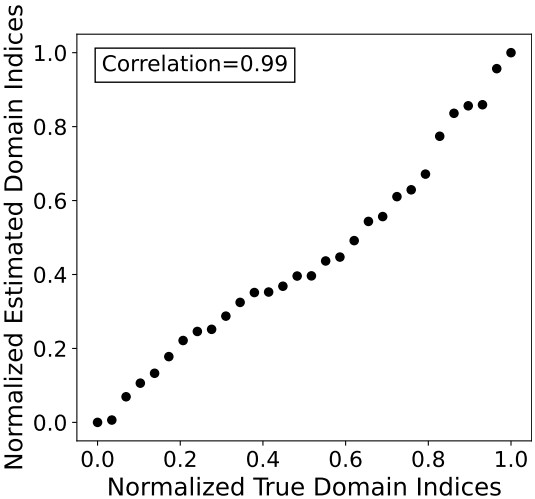

Figure 8: Inferred domain indices (reduced to 1 dimension by PCA) with true domain indices for dataset *Circle*. GMDI's inferred indices have a correlation of 0.99 with true indices, even though *GMDI does not have access to true indices during training*.

## E  Architecture and Implementation Details

### E.1  Architecture

The latent variables are estimated by neural networks. Specifically, for the local domain index $\boldsymbol{u}$, we employ ResNet-18 [9] to approximate its posterior on *CompCars* dataset, while using multi-layer perceptrons for the other datasets. Additionally, all neural networks are implemented as multi-layer perceptrons. We use the features obtained from the ResNet-18 as inputs to our model. Furthermore, all inputs are uniformly normalized.

We implement our model based on the code of VDI[41]. We appreciate the authors for making their code publicly available. We run experiments on a single machine using 1 NVIDIA GeForce RTX 2080Ti with 11GB memory, 56 Intel Xeon CPUs (E5-2680 v4 @ 2.40GHz). It takes about 30 minutes to train GMDI with K=2 on synthetic datasets , 4 hours on *TPT-48*, and 6 hours on *CompCars* with K=3.

## E.2 Hyperparameters

We set the maximum number of mixture components $K$ from 2,3, and the concentration parameter $\alpha$ to 1 throughout the experiments. Except for *DG-15* and *DG-60* datasets, which have a batch size of 32, all other datasets use a batch size of 16. Our model is trained with 20 to 100 warmup steps, learning rates ranging from $1 \times 10^{-5}$ to $1 \times 10^{-3}$, and $\lambda$ ranging from 0.1 to 1.

## F  Broader Impacts

Our model has the potential to be applied to various domain shift problems, which also implies the possibility of unintended negative consequences. However, we have not identified any specific societal harms associated with our model. If used maliciously, it could lead to negative impacts.

## G  Pseudo Codes

The procedure of our proposed model GMDI is summarized by the pesuedo codes in Algorithm 1. Let $\varphi$ represent the parameters of the distribution $p(\cdot)$ in the generative process.

---

**Algorithm 1** Bayesian Domain Adaptation with Gaussian Mixture Domain-Indexing

---

**Input:** Dataset $\mathcal{D}^S$ and $\mathcal{D}^T$, maximum number of mixture components $K$, concentration parameter $\alpha$, learning rate $\zeta$.

1: Initialize parameters: $\boldsymbol{\varphi}, \boldsymbol{\psi} = \{\boldsymbol{\psi}_u, \boldsymbol{\psi}_\theta, \boldsymbol{\psi}_z\}; \boldsymbol{\eta}_k, \forall k = 1, ..., K;$
2: **repeat**
3:     Update $\boldsymbol{\gamma}_k$ with $\boldsymbol{\gamma}_{k,1} = 1 + \boldsymbol{\eta}_k$ and $\boldsymbol{\gamma}_{k,2} = \alpha + \sum_{i=k+1}^{K} \boldsymbol{\eta}_i, \forall k = 1, ..., K;$
4:     Update $\boldsymbol{\eta}$ with Equation 37;
5:     Sample $\boldsymbol{u}$ from Equation39;
6:     Sample $\boldsymbol{\theta}$ based on Equation38;
7:     Sample $\boldsymbol{z}$ according to Equation 40;
8:     Update $\boldsymbol{\varphi} \leftarrow \boldsymbol{\varphi} + \zeta \nabla_{\boldsymbol{\varphi}} \mathcal{L}_{\text{GMDI}}, \boldsymbol{\psi} \leftarrow \boldsymbol{\psi} + \zeta \nabla_{\boldsymbol{\psi}} \mathcal{L}_{\text{GMDI}}$ .
9: **until** converge

---

## H  Dataset Summary

Table 3 [41] summarizes the statistics for all the datasets used in our experiments.

Table 3: Summary of statistics and settings in different datasets.

| Dataset | Numbers of samples | Input dim | Synthetic/Real | Task |
|---------|--------------------|-----------|----------------|------|
| *Circle* | 3,000 | 2 | Synthetic | 2-Way classification |
| *DG-15* | 1,500 | 2 | Synthetic | 2-Way classification |
| *DG-60* | 6,000 | 2 | Synthetic | 2-Way classification |
| *TPT-48* | 6,912 | 6 | Real | Regression |
| *CompCars* | 18,735 | $224 \times 224$ | Real | 4-Way classification |

## I  Dataset

### I.1  Circle

Figure 9 [41] visualizes the detailed information of *Circle* dataset. It contains 30 domains and is used for binary classification task. The data points in the *Circle* are arranged in a semicircular shape, with each domain occupying a different section of the semicircle. There is a decision boundary that separates the different labels. We use the first six domains as the source domains and the remaining 24 domains as the target domains.

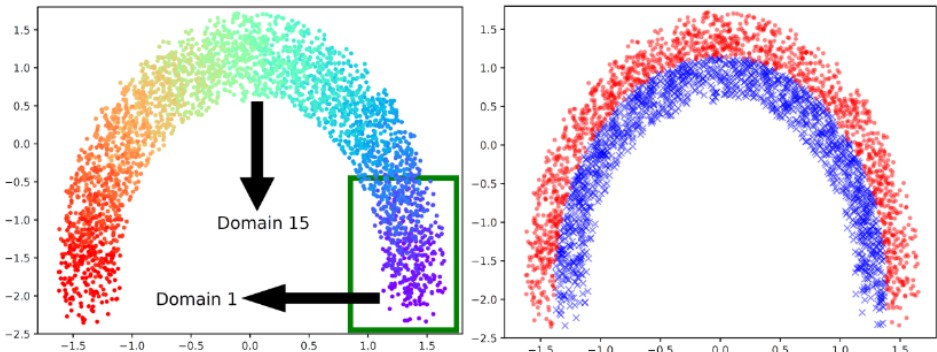

Figure 9: The Circle dataset with 30 domains. **Left**: Different colors indicate ground-truth domain indices. The first 6 domains (in the green box) are source domains. **Right**: Ground-truth labels for *Circle*, with red dots and blue crosses as positive and negative data points, respectively.

## I.2 DG-15 and DG-60

*DG-15 and DG-60* datasets containing 15 and 60 domains, respectively. Both used for binary classification task. Adjacent domains in the datasets have similar decision boundaries. For these two datasets, we select 6 connected domains as the source domains, while the remaining domains serve as the target domains.

## I.3 TPT-48

Figure 10 [40] visualizes the detailed information of *TPT-48* dataset. It contains the monthly average temperatures of 48 contiguous states in the United States from 2008 to 2019. Our regression task is to predict the average temperatures for the next 6 months using the average temperatures of the first 6 months. We divide this task into two finer-grained regression tasks with different adaptation directions:

- W (6) $\rightarrow$ E (42): Adapting models from the 6 states in the west to the 42 states in the east.

- N (24) $\rightarrow$ S (24): Adapting models from the 24 states in the north to the 24 states in the south.

To better verify performance, the target domains in the above two regression tasks are divided into three groups based on their distance from the closest source domains. *level-1 target domains*: one hop away from the closest source domain. *level-2 target domains*: two hops away from the closest source domain. *level-3 target domains*: more than two hops away from the closest source domain.

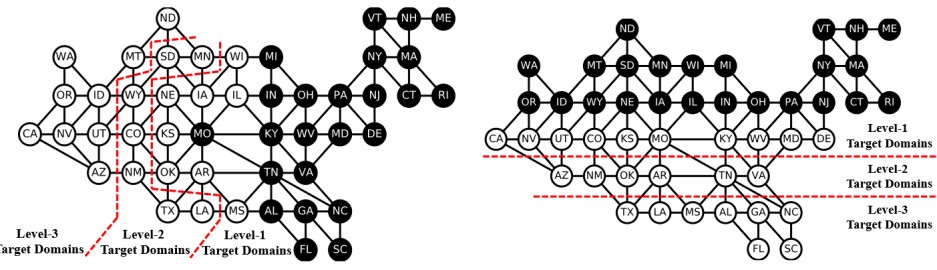

Figure 10: Domain graphs for the two adaptation tasks on TPT-48, with black nodes indicating source domains and white nodes indicating target domains. **Left**: Adaptation from the 24 states in the east to the 24 states in the west. **Right**: Adaptation from the 24 states in the north to the 24 states in the south.

### I.4 CompCars

*CompCars* dataset includes three attributes: car types, viewpoints, and years of manufacture (YOMs). We select a subset of *CompCars* with 4 types (MPV, SUV, sedan and hatchback), 5 viewpoints (front(F), rear (R), side (S), front-side (FS), and rear-side (RS)), and 6 YOMs(2009, 2010, 2011, 2012, 2013, 2014) for our experiments. This subset is divided into 30 domains(5 viewpoints × 6 YOMs) based on viewpoints and YOMs. The car types are used as labels for prediction. The first domain, which has the front view and YOM 2009, is treated as the source domain, while the remaining 29 domains are target domains.

## J   Notation

Table 4 provides a summary of the notations used in this paper.

Table 4: Summary of notations.

| Symbol | Definition |
|:---:|:---|
| $\boldsymbol{x}$ | Input data |
| $y$ | Label of data |
| $w$ | Domain identity |
| $\boldsymbol{\theta}$ | Global domain index |
| $\boldsymbol{u}$ | Local domain index |
| $\boldsymbol{z}$ | Data encoding |
| $v$ | Latent categorical variable |
| $\varepsilon$ | Parameter for prior probability distribution of domain index |
| $\alpha$ | Concentration parameter of the CRP |
| $\boldsymbol{\beta}$ | independent random variable with a Beta distribution in the stick-breaking representation |
| $\boldsymbol{\pi}$ | Probability vector in the stick-breaking |
| $\boldsymbol{\gamma}$ | Parameter of $q(\boldsymbol{\beta})$ |
| $\boldsymbol{\eta}$ | Parameter of $q(v)$ |
| $\boldsymbol{\psi}_u$ | Parameter of $q(\boldsymbol{u} \mid \boldsymbol{x})$ |
| $\boldsymbol{\psi}_\theta$ | Parameter of $q(\boldsymbol{\theta} \mid \boldsymbol{u})$ |
| $\boldsymbol{\psi}_z$ | Parameter of $q(\boldsymbol{z} \mid \boldsymbol{x}, \boldsymbol{u}, \boldsymbol{\theta}^v)$ |
| $\boldsymbol{\phi}$ | Parameter of $q(\boldsymbol{u}, \boldsymbol{\theta}^v, \boldsymbol{z}|\boldsymbol{x})$ |
| $\boldsymbol{\xi}$ | Parameter of $q(\boldsymbol{u}, \boldsymbol{\theta}^v|\boldsymbol{x})$ |
| $\lambda$ | Hyper-parameter that balances two terms in objective function |

## K   Additional Experimental Results

To evaluate the impact and computational cost of the CRP, we conduct ablation and computational cost experiments. We implement GMDI w/o CRP using Gumbel-Softmax [11]. The number of components for GMDI w/o CRP is set to the upper bound $K$ of GMDI, and the hyperparameter temperature $\tau$ for Gumbel-Softmax ranges from 0.1 to 50 (with the best performance reported). "Total time" refers to the total training duration, which concludes when the loss converges.

Table 5: The results of the ablation and computational cost experiments on *TPT-48*.

| Task | Method | MSE | Total time | Epochs | Time per epoch |
|:---:|:---:|:---:|:---:|:---:|:---:|
| | VDI | 2.496 | 1h 24m 18s | 400 | 13s |
| W → E | GMDI w/o CRP | 2.471 | 2h 14m 54s | 500 | 16s |
| | GMDI | 2.087 | 1h 31m 13s | 300 | 18s |
| | VDI | 3.160 | 1h 51m 43s | 500 | 13s |
| N → S | GMDI w/o CRP | 3.050 | 2h 22m 58s | 500 | 17s |
| | GMDI | 2.493 | 2h 2m 35s | 400 | 18s |

Table 6: The results of the ablation and computational cost experiments on *CompCars*.

| Method | Accuracy(%) | Total time | Epochs | Time per epoch |
|--------|-------------|------------|--------|----------------|
| VDI | 42.5 | 3h 13m 15s | 600 | 19s |
| GMDI w/o CRP | 43.0 | 4h 3m 48s | 700 | 21s |
| GMDI | 44.4 | 4h 16m 14s | 600 | 26s |

The experimental results are shown in Table 5 and Table 6. We find that although the proposed GMDI has a longer "Time per epoch" compared to GMDI w/o CRP, it converges faster due to the flexible number of components adaptively controlled by CRP. Therefore, the "Total time" is roughly the same as GMDI w/o CRP. On the *TPT-48*(W->E) dataset, due to faster convergence, the "Total time" of GMDI is less than that of GMDI w/o CRP and is even comparable to VDI. In all three datasets, the performance of GMDI w/o CRP is worse than that of GMDI. On the two *TPT-48* datasets, compared to the baseline VDI, GMDI w/o CRP reduces MSE by 1% and 3%, whereas GMDI reduces MSE by 16% and 21%, surpassing GMDI w/o CRP. On the *CompCars* dataset, GMDI's accuracy is higher than that of GMDI w/o CRP. These results indicate that although using a fixed-component GMM is simpler, the computational costs are roughly equivalent to using CRP, but the performance is inferior to CRP, demonstrating the significance of CRP in GMDI.

Additionally, compared to VDI, which models the domain index as a single Gaussian distribution, GMDI's computational costs are only slightly higher, yet its performance is superior. For large-scale datasets with numerous domains, modeling the domain index as a simple single Gaussian distribution may result in poor performance due to the dataset's complexity. The experimental results indicate that GMDI has broad applicability.

