# OpenReview forum: "Bayesian Domain Adaptation with Gaussian Mixture Domain-Indexing"
_NeurIPS.cc/2024/Conference — NeurIPS 2024 poster_

### Official Review · Reviewer_72He · 2024-07-11

**Soundness:** 3
**Presentation:** 3
**Contribution:** 3
**Rating:** 6
**Confidence:** 4

**Summary:**

The paper proposes a novel method, "Gaussian Mixture Domain-Indexing" (GMDI), to address domain adaptation with inaccessible domain indices. The technique improves upon prior work by modeling the domain indices prior with a Gaussian Mixture. Empirically, it has been shown that the proposed method achieves state-of-the-art performance in classification and regression tasks.

**Strengths:**

**Novelty**: The paper proposes a novel technique to address the issue of domains in domain adaptation having multiple semantics. The method is a natural extension from prior work (VDI) by changing the Gaussian prior to a Gaussian Mixture.

**Theoretical Support**: The paper proves the correctness of the proposed method with solid theoretical derivations.

**Empirical Validation**: Extensive experiments show that GMDI significantly outperforms state-of-the-art methods in both classification and regression tasks, with substantial improvements in accuracy and mean squared error (MSE). I particularly like the illustration of the learned domain indices. Figures 4 and 5 clearly show that GMDI learns the domain indices more accurately compared to the prior state-of-the-art method, VDI. Good job!

**Weaknesses:**

**Clarity**: Some equations in the paper might have typos. Please see the details in the next section.

**Computational Overhead**: The use of CRP and dynamic mixtures increases computational overhead, which might make the method less practical for large-scale or real-time applications without further optimization. It would be helpful if the authors could provide some comments and empirical analysis on the computational overhead of GMDI.

**Questions:**

**Typos?** Should the left side of Equation (1) be $p(z|x,\epsilon)$ or should $\int_{x}$ be added to the right side? A similar issue seems to exist in Equation (5) as well.

**Abstract Claim** I do not understand the authors' claim in the abstract, “For classification, GMDI improves accuracy by at least 63% (from 33.5% to 96.5%).” It seems from Table 1, the result for DG-60, the authors picked the worst performance of all baselines, ADDA with an accuracy of 33.5%. It is unclear why the authors claim the improvement is at least 63%.

**Limitations:**

I do not see specific limitations that might lead to potential negative societal impact.

---

> ### Author Rebuttal · Authors · 2024-08-07
>
> We thank the reviewer for the detailed and positive comments. The following is our responses to the questions mentioned in the comments.
>
> **1. Computational Overhead: The use of CRP and dynamic mixtures increases computational overhead, which might make the method less practical for large-scale or real-time applications without further optimization. It would be helpful if the authors could provide some comments and empirical analysis on the computational overhead of GMDI.**
>
> We appreciate this useful suggestion of the reviewer. In the global "Author Rebuttal", we conduct detailed ablation and  computational overhead experiments comparing the proposed GMDI with GMDI w/o CRP. Please kindly refer to the global "Author Rebuttal".
>
> **2. Typos? Should the left side of Equation (1) be $p\left ( z\mid x, \varepsilon   \right )$ or should $\int_{x}$ be added to the right side? A similar issue seems to exist in Equation (5) as well.**
>
> Thanks for pointing out the typos. The left side of Equation (1) and Equation (5) should be corrected to $p\left ( z\mid x, \varepsilon \right )$. We will include the correction  in the final version.
>
> **3. Abstract Claim I do not understand the authors' claim in the abstract, “For classification, GMDI improves accuracy by at least 63% (from 33.5% to 96.5%).” It seems from Table 1, the result for DG-60, the authors picked the worst performance of all baselines, ADDA with an accuracy of 33.5%. It is unclear why the authors claim the improvement is at least 63%.**
>
> We apologize for the misleading claim. We will correct it in the final manuscript.

---

> > ### Comment · Reviewer_72He · 2024-08-12
> > **Official Comment by Reviewer 72He**
> >
> > Thank you. My concerns are well addressed. I stay postive for this manuscript.

---

> > > ### Author Response · Authors · 2024-08-12
> > > **Thanks.**
> > >
> > > Thank you very much for your positive support.

---

### Official Review · Reviewer_TJok · 2024-07-12

**Soundness:** 3
**Presentation:** 3
**Contribution:** 3
**Rating:** 7
**Confidence:** 4

**Summary:**

This paper proposes a Bayesian Domain Adaptation method with Gaussian Mixture Domain-Indexing (GMDI) to address the challenge of inferring domain indices when they are unavailable. Existing methods often assume a single Gaussian prior for domain indices, ignoring the inherent structures among domains. GMDI models domain indices as a mixture of Gaussian distributions, with the number of components dynamically determined by the Chinese Restaurant Process. This approach provides a higher level of flexibility and effectiveness in adapting to diverse target domains. Theoretical analysis demonstrates that GMDI achieves a more stringent evidence lower bound, closer to the log-likelihood. Extensive experiments on classification and regression tasks show that GMDI significantly outperforms baselines, achieving state-of-the-art performance.

**Strengths:**

1. GMDI is the first to model domain indices as a mixture of Gaussian distributions, allowing it to capture the inherent structures among different domains. This approach provides a more flexible and powerful way to infer domain indices.

2. By using the Chinese Restaurant Process, GMDI can dynamically determine the number of mixture components, adapting to varying numbers of domains. This enhances its capability to handle complex datasets with an unknown number of domains.

3. The paper provides a detailed theoretical analysis, demonstrating that GMDI achieves a more stringent evidence lower bound and a tighter upper bound of the objective function compared to existing methods. This theoretical foundation supports the effectiveness of the proposed approach.

**Weaknesses:**

1. GMDI relies on the availability of domain identities but cannot infer them as latent variables. This limits its applicability to scenarios where domain identities are also unknown.

2. The use of the Chinese Restaurant Process and Gaussian Mixture Model can be computationally intensive, especially for large-scale datasets with numerous domains. This could hinder the scalability of GMDI.

3. In the experiment, the binary classification task is not challenging, which makes the performance advantage not convincing. The multi-classification tasks are necessary.

**Questions:**

See weaknesses.

**Limitations:**

See weaknesses.

---

> ### Author Rebuttal · Authors · 2024-08-07
>
> We thank the reviewer for the positive comments, the insightful questions, and helpful suggestions. The following are our responses to the questions mentioned in the comments.
>
> **1. GMDI relies on the availability of domain identities but cannot infer them as latent variables. This limits its applicability to scenarios where domain identities are also unknown.**
>
> Thank you to the reviewer for pointing out this weakness. Our proposed GMDI focus on inferring domain indices following a mixture of Gaussian distributions, with the number of mixture components dynamically determined by a Chinese Restaurant Process (CRP). Extensive experiments on classification and regression tasks demonstrate the strong domain index modeling capability of GMDI, significantly outperforming state-of-the-art methods. However, we acknowledge that GMDI's reliance on domain identities limits its applicability in situations where domain identities are unknown. We plan to address this issue in future work.
>
> **2. The use of the Chinese Restaurant Process and Gaussian Mixture Model can be computationally intensive, especially for large-scale datasets with numerous domains. This could hinder the scalability of GMDI.**
>
> We appreciate the reviewer for highlighting this issue. In the global "Author Rebuttal", we conduct detailed ablation and computational cost experiments comparing the proposed GMDI with GMDI w/o CRP and VDI.  Please kindly refer to the global "Author Rebuttal" for a comprehensive analysis.
>
> **3. In the experiment, the binary classification task is not challenging, which makes the performance advantage not convincing. The multi-classification tasks are necessary.**
>
> Thank you for bringing this to our attention. Fllowing VDI, we have tested the performance of GMDI on the multi-classification dataset *CompCars*. The *CompCars* dataset is a 4-way classification dataset containing 18,735 samples across 30 domains. Due to the scarcity of datasets with ground-truth domain indices, we leave the extension of experiments on new datasets for future research. More details about the datasets used are available in Appendix H of the paper.

---

> > ### Comment · Reviewer_TJok · 2024-08-12
> >
> > Thank you. The authors have addressed all my concerns.

---

> > > ### Author Response · Authors · 2024-08-12
> > > **Thanks**
> > >
> > > Thank you very much for your positive support.

---

### Official Review · Reviewer_uzxH · 2024-07-12

**Soundness:** 2
**Presentation:** 2
**Contribution:** 2
**Rating:** 4
**Confidence:** 4

**Summary:**

The paper introduces the Gaussian Mixture Domain-Indexing (GMDI) algorithm for domain adaptation when domain indices are unavailable. Unlike traditional methods that use a simple Gaussian prior, GMDI employs a Gaussian Mixture Model adjusted by a Chinese Restaurant Process, enabling adaptive determination of mixture components.

**Strengths:**

- The use of a Gaussian mixture to represent complex domain indices seems interesting.
- The proposed method demonstrates superior performance over baseline models in experimental results.
- The technical part (the proposed method) seems non-trivial and its complexity seems sufficient, yet it might not be necessary.

**Weaknesses:**

- The learning loss for the proposed model is over too complex, featuring multiple conditional Kullback-Leibler divergences, which might complicate implementation and interpretation.
- The paper lacks clear definitions for the 'local' and 'global' domain index within the probabilistic graphical model, which could confuse readers about the model's scope and applicability.
- While introducing the Chinese Restaurant Process (CRP) adds flexibility to the Gaussian Mixture Model, it also increases computational costs. For the problem described in Figure 1, a fixed-component GMM might have been a simpler and more effective solution, though the number of components may not be that flexible.
- The authors should also seriously consider empirically comparing the proposed method with a fixed-component counterpart, e.g. by simply using Gumbol softmax to infer the Gaussian component.
- The related work may not be sufficiently discussed. For example [a] discussed an end-to-end approach that learns the domain index using adversarial learning; [b] takes the domain index/identity as a latent dynamical system, coupled with adversarial learning.

[a] Out-of-distribution Representation Learning for Time Series Classification
[b] Extrapolative Continuous-time Bayesian Neural Network for Fast Training-free Test-time Adaptation

----------------Minor------------------

- The connection between Equation (5) and Equation (6) is not clearly explained, leaving a gap in understanding the sequential logic of the model's formulation.
-The paper's clarity and accuracy in writing could be improved. For instance, the statement "DP requires a predefined number of components" is misleading, as Dirichlet Processes are inherently nonparametric and do not require a predefined number of components.
- Several symbols used in the equations are not adequately explained (both in the major paper and appendix), making it difficult to fully grasp the proposed model and its mathematical foundations.
- The proofs and the theoretical part seem to closely follow that of VDI, making it hard to evaluate its novelty.

**Questions:**

See weaknesses.

**Limitations:**

The limitations are  addressed in the Sec. conclusion.

---

> ### Author Rebuttal · Authors · 2024-08-07
>
> We thank the reviewer for the detailed comments and constructive suggestions. The following is our responses to the questions mentioned in the comments.
>
> **1. The learning loss for the proposed model is over too complex, featuring multiple conditional Kullback-Leibler divergences, which might complicate implementation and interpretation.**
>
> Thanks for pointing out this shortcoming. The learning loss for the proposed GMDI is indeed somewhat complex. GMDI follows the framework of VDI, which involves multiple latent variables and requires KL divergence constraints on these variables, making this complexity unavoidable. However, by modeling the domain index as a mixture of Gaussian distributions and using the Chinese Restaurant Process (CRP), GMDI demonstrates excellent performance on the relatively more complex *CompCars* dataset. This indicates its potential for application to even more intricate datasets, showcasing its broad applicability.
>
> **2. The paper lacks clear definitions for the 'local' and 'global' domain index within the probabilistic graphical model, which could confuse readers about the model's scope and applicability.**
>
> We apologize for the confusion. GMDI follows the setting of VDI.
>
> (1) Local domain index $u$: It contains instance-level information, meaning each data point has a unique local domain index. Essentially, the local domain index can be viewed as an intermediate latent variable.
>
> (2) Global domain index $\theta$: It contains domain-level information, meaning all data points within the same domain share the same global domain index. The global domain index is the true "domain index".
>
> Our proposed GMDI models the global domain index (which contains domain-level information) as a mixture of Gaussian distributions. This mixture provides a higher level of flexibility in a larger latent space, thereby enhancing performance. We will incorporate the above definitions into the final version.
>
> **3. While introducing the Chinese Restaurant Process (CRP) adds flexibility to the Gaussian Mixture Model, it also increases computational costs. For the problem described in Figure 1, a fixed-component GMM might have been a simpler and more effective solution, though the number of components may not be that flexible.**
>
> We appreciate the reviewer for highlighting this issue. In the global "Author Rebuttal", we conduct detailed ablation and computational cost experiments comparing the proposed method with a fixed-component GMM. Please kindly refer to the global "Author Rebuttal".
>
> **4. The authors should also seriously consider empirically comparing the proposed method with a fixed-component counterpart, e.g. by simply using Gumbol softmax to infer the Gaussian component.**
>
> Thank you for the insightful suggestion. In the global "Author Rebuttal", we conduct detailed ablation and computational cost experiments comparing the proposed method with a fixed-component counterpart using Gumbol softmax. Please kindly refer to the global "Author Rebuttal".
>
> **5. The related work may not be sufficiently discussed. For example [a] discussed an end-to-end approach that learns the domain index using adversarial learning; [b] takes the domain index/identity as a latent dynamical system, coupled with adversarial learning.**
>
> Thanks for the mention of these two related works. We follow the setting of VDI, where the concepts of "domain index" and "domain identity" indeed share significant similarities, though they are somewhat different. The two mentioned papers primarily focus on "domain identity", and they are both highly valuable and worthy of discussion. We will discuss them in the related work section of the final version.
>
> ----------Minor----------
>
> **6. The connection between Equation (5) and Equation (6) is not clearly explained, leaving a gap in understanding the sequential logic of the model's formulation. -The paper's clarity and accuracy in writing could be improved. For instance, the statement "DP requires a predefined number of components" is misleading, as Dirichlet Processes are inherently nonparametric and do not require a predefined number of components.**
>
> We appreciate the reviewer for pointing out the issues.
>
> (1) Equation 5 and Equation 6 both represent the posterior factorization of data encoding $z$. $\theta$ in Equation 5 is the global domain index. While GMDI models the global domain index $\theta$ as a mixture of Gaussian distributions, $\theta ^{v}$ in Equation 6 indicates the $v$-th component of the mixture distribution of $\theta$ with the prior $p\left ( v  \right )$.
>
> (2) Thanks for pointing out the misleading statement. The Dirichlet Process (DP) is indeed non-parametric. Due to the difficulty of direct construction, we apply the Chinese Restaurant Process by stick-breaking with a predefined bound $K$ to implement the DP. We will correct it with the explanatory sentences above.
>
> **7. Several symbols used in the equations are not adequately explained (both in the major paper and appendix), making it difficult to fully grasp the proposed model and its mathematical foundations.**
>
> We apologize for the missing explanations of some notations. Below is a detailed explanation of the missing notation：
>
> $\beta$: independent random variable with a Beta distribution in the stick-breaking representation;
>
> $\pi$: the probability vector in the stick-breaking;
>
> $\gamma$: parameter of $q(\beta)$;
>
> $\eta$: parameter of $q(v)$.
>
> We will include the explanations of the above notations in the final version.
>
> **8. The proofs and the theoretical part seem to closely follow that of VDI, making it hard to evaluate its novelty.**
>
> Thank you for bringing this to our attention. Compared to VDI, the theoretical contribution of the proposed GMDI is that it proves GMDI achieves a more stringent evidence lower bound, as well as a tighter upper bound of the objective.  Theoretically, it demonstrates that GMDI can effectively infer optimal domain indices.

---

> > ### Comment · Reviewer_uzxH · 2024-08-09
> > **Thanks**
> >
> > Thanks for the efforts. The author's response addressed some of my concerns, but my major concerns about using a simpler solution such as fixed component GMM still exists. Though the additional results are provided, I am surprised that GMDI w/o CRP fails to beat VDI in terms of accuracy, while haveing a smaller MSE. This could due to overfiting. Based on the general quality of the paper and above concerns, i decide to downgrades my ratings.

---

> > > ### Author Response · Authors · 2024-08-09
> > > **Incorrect comments and unfair downgraded rating from the reviewer uzxH**
> > >
> > > Thanks for your response. However, we are quite sure that your following claim is not correct, "I am surprised that GMDI w/o CRP fails to beat VDI in terms of accuracy." In fact, on the *CompCars* dataset, GMDI w/o CRP achieves an accuracy of 43.0%, while the accuracy of VDI is 42.5%. Please see the corresponding details in the global "Author Rebuttal". This indicates that GMDI w/o CRP indeed performs better than VDI, contrary to your claim. We have conducted detailed experiments on *TPT-48* dataset for regression task and *CompCars* dataset for classification task. On all datasets, GMDI w/o CRP slightly outperforms VDI, while it lags far behind GMDI. Because of the above, the downgraded rating by you is unfair for us. Thank you very much and looking forward to your positive response.

---

> > > > ### Comment · Reviewer_uzxH · 2024-08-09
> > > >
> > > > It could be my mistake about the results. Since the performance gap of these two models are relatively small. Can the authors further provide significant testing results?

---

> ### Author Response · Authors · 2024-08-10
> **Thanks for the prompt response**
>
> Thank you very much for your prompt response. The performance gap between the two models, i.e., GMDI w/o CRP and  VDI, is significant, which is what we expected and good result. As you suggested, we further applied a two-tailed paired t-test and found that the statistical significance of the observed performance difference between these two models is significant for alpha=0.05. Because of the positive experimental result, we are expecting you to at least upgrade the current rating score to the original rating score. Thank you very much for you kind support.

---

> > ### Comment · Reviewer_uzxH · 2024-08-11
> >
> > Please let us know about the search space of the hyperparameter K for each dataset.  Is K=1 for CompCars?  If so, it is no wonder the results are not significantly different from those of VDI. Furthermore, the search space of K should be sufficiently large for GMM-based variational inference models.  For now, I still have concerns about the over-complexity of the method. Such complexity may not be necessary for considering the multimodal assumption of the domain index. Therefore, I decided to keep my current ratings.

---

> > > ### Author Response · Authors · 2024-08-12
> > > **Thanks for the prompt response**
> > >
> > > Thank you for your response. The search space of the hyperparameter K for each dataset ranges from 1 to 5. Due to limitations in computational resources, we did not perform a broader search for the hyperparameter K. It's clear that the hyperparameter K for the *CompCars* dataset is not 1. We have already stated in the global "Author Rebuttal" that "The number of components for GMDI w/o CRP is set to the upper bound K of GMDI," and we have detailed the upper bound K for each dataset in the Experimental Study section of the paper. For the *CompCars* dataset, the hyperparameter K is 3, not 1. In our previous response, based on your suggestion, we have further applied a two-tailed paired t-test and found that the statistical significance of the observed performance difference between GMDI w/o CRP and VDI is significant for alpha=0.05. We are puzzled by your claim in the review that "the results are not significantly different from those of VDI," as this contradicts the results of our significance testing.
> > > Moreover, while the search space of K indeed needs to be sufficiently large for GMM-based variational inference models, this is actually where GMDI shines. Using a fixed-component GMM requires extensive searching for K, a process that is very time-consuming. In contrast, GMDI leverages CRP to dynamically adjust the number of components with just an simple upper bound on K. Although the use of CRP in GMDI is more complex compared to fixed-component GMM, CRP's dynamic adaptability significantly reduces the cost of searching for K, making the use of CRP essential.
> > >
> > > You downgraded the rating because of the incorrect statement "I am surprised that GMDI w/o CRP fails to beat VDI in terms of accuracy.". We have addressed this issue raised by you at the first-round discussion (see the previous discussion). And now we believe we also have addressed the issue raised at the second-round discussion (the above discussion). Because the issues have been addressed appropriately,  we wish you are able to keep at least the original rating for our paper rather than the current downgraded score. Thank you very much for your positive support.

---

> > > > ### Comment · Reviewer_uzxH · 2024-08-12
> > > >
> > > > I am sorry that I still decided to keep my current ratings. My major concerns are the following: 1 current solution is over too complex, which is not necessary. 2 The baselines of fixed component GMM are not fully investigated or tuned tuned (e.g. hyperparameters of gumbol softmax version and other GMM based solution e.g.  GMVAE), making the introduction of CRP not well-motivated.

---

> > > > > ### Author Response · Authors · 2024-08-12
> > > > > **Arbitrary, reckless, and vague reasons for downgrading the rating from Reviewer uzxH.**
> > > > >
> > > > > The reasons given by the reviewer for downgrading the rating are very arbitrary, reckless, and vague.
> > > > >
> > > > > During the first-round discussion in the discussion session, the claim made by the reviewer in response to our "Rebuttal"—"I am surprised that GMDI w/o CRP fails to beat VDI in terms of accuracy"—was absolutely incorrect. The reviewer hastily downgraded his/ her rating due to this basic mistake (misreading the results in the table).
> > > > >
> > > > > In the second-round discussion, although we have addressed the claim raised in the first round and the reviewer admitted he/ she  has made a mistake (“It could be my mistake about the results”), he/ she raised a new but inessential issue: the reviewer guessed that "K=1 for CompCars," and based on this assumption, he/ she still maintained the downgraded rating. However, we clearly stated that K=3 for CompCars, not K=1.
> > > > >
> > > > > In the third-round discussion, the reviewer stated that "The baselines of fixed-component GMM are not fully investigated or tuned (e.g., hyperparameters of the Gumbel-Softmax version and other GMM-based solutions like GMVAE)." However, we have already thoroughly tuned the hyperparameters of the Gumbel-Softmax version (i.e., GMDI w/o CRP). Additionally, it is unrealistic and unnecessary to compare all variants of fixed-component GMMs within a short time frame. We have already conducted comparative experiments with the Gumbel-Softmax version (i.e., GMDI w/o CRP). Below are our responses to the concerns mentioned in the third-round discussion.
> > > > >
> > > > > **1. "The current solution is overly complex, which is not necessary."**
> > > > >
> > > > > We have already addressed this issue in our previous response. The use of CRP in our proposed GMDI is actually essential. We model the domain index as a mixture of Gaussians, but the number of mixture components for the domain index is unknown beforehand. CRP allows us to adaptively determine the number of mixture components. Without CRP, the number of mixture components for the domain index cannot be determined, so while CRP slightly increases the complexity, it is absolutely necessary.
> > > > >
> > > > > **2. "The baselines of fixed-component GMM are not fully investigated or tuned (e.g., hyperparameters of the Gumbel-Softmax version and other GMM-based solutions like GMVAE), making the introduction of CRP not well-motivated."**
> > > > >
> > > > > In our previous responses, we have explained that the hyperparameter K for the Gumbel-Softmax version (i.e., GMDI w/o CRP) ranges from 1 to 5. Additionally, in the global "Author Rebuttal," we stated that the hyperparameter $\tau$ ranges from 0.1 to 50. We have indeed searched for the hyperparameters of the Gumbel-Softmax version (i.e., GMDI w/o CRP) within a sufficiently large space. The experimental results show that our proposed GMDI has a clear advantage over the Gumbel-Softmax version (i.e., GMDI w/o CRP). The motivation for using CRP in GMDI is to adaptively determine the number of mixture components for the domain index. Since the number of mixture components for the domain index is unknown beforehand, the Gumbel-Softmax version (i.e., GMDI w/o CRP) requires constant tuning of the K value, which is extremely time-consuming and not suitable for domain index applications. In contrast, CRP with a simple upper bound on K can adaptively determine the number of mixture components. The GMVAE mentioned by the reviewer also requires constant tuning of the K value, and all fixed-component GMMs, in principle, cannot adaptively determine the number of mixture components. It is unrealistic and unnecessary to conduct comparative experiments with all fixed-component GMM variants within a short timeframe. Furthermore, we have emphasized the motivation for applying CRP in GMDI in the paper.
> > > > >
> > > > > We believe that we have addressed all the issues raised by the reviewer during the discussion. We hope that the reviewer's evaluation is based on a careful review of the paper and rebuttal, rather than on random, reckless, and vague reasons for rejection and downgrading.

---

> > > > > > ### Comment · Area_Chair_HVnC · 2024-08-12
> > > > > >
> > > > > > Dear Reviewer uzxH,
> > > > > >
> > > > > > It seems the authors have clarified some of the potential misunderstandings. Could you please take a look to see if these clarifications change your mind?
> > > > > >
> > > > > > Best,
> > > > > >
> > > > > > AC

---

> > > > > > ### Comment · Reviewer_uzxH · 2024-08-13
> > > > > >
> > > > > > Generally speaking, this paper focuses on an interesting issue within VDI, specifically its neglect of the multimodal nature of the latent index. However, the study primarily builds upon the existing framework of VDI, and its theoretical developments, which are closely aligned with those of NDI, are mostly relegated to the appendix without appropriate citations. This lack of clarity suggests that some key proofs might be incorrectly attributed to this work.
> > > > > >
> > > > > > Additionally, the use of the Chinese Restaurant Process (CRP) to manage an unknown number of Gaussian components is not innovative, as thoroughly discussed in prior research [1]. Therefore, both the technical novelty and theoretical contributions of this paper seem incremental.
> > > > > >
> > > > > > The experiments presented are not sufficiently comprehensive. I can understand that the authors have difficulties in conducting experiments using additional GMM variants, yet such comparisons are essential for a robust evaluation. Notably in the experiments, the CRP model frequently identifies the optimal number of components as two or three. It is important to figure out why CRP outperforms fixed-component GMMs, given that the "optimal"  number of Gaussian components is known.A more thorough ablation study could clarify these results.
> > > > > >
> > > > > > Furthermore, the methodology should be tested on data featuring more complex multimodal distributions, which may require a larger number of Gaussian components for accurate representation of the index. Such evaluation would help validate the method’s effectiveness and demonstrate more advantages over a fixed components method.
> > > > > >
> > > > > > Given the above concerns, I would lean towards rejecting this paper, as it requires major revisions.
> > > > > >
> > > > > > [1] Li, N., Li, W., Jiang, Y., & Xia, S. T. (2022, August). Deep Dirichlet process mixture models. In Uncertainty in Artificial Intelligence (pp. 1138-1147). PMLR.

---

> ### Author Response · Authors · 2024-08-10
> **Further explanation of the experimental results**
>
> To avoid misunderstanding and further clarify the experimental results presented in the global "Author Rebuttal," we would like to provide the following additional explanation. VDI is our baseline, which models the domain index as a single Gaussian. Our proposed GMDI models the domain index as a mixture of Gaussians and incorporates CRP. GMDI w/o CRP is a version that follows your suggestion by not using CRP and instead utilizing a fixed-component GMM (Gaussian Mixture Model with a fixed number of components).
>
> We conducted detailed experiments on the relatively complex *TPT-48* and *CompCars* datasets. The results show that GMDI w/o CRP performs better than VDI but falls short of GMDI. This indicates that using a fixed-component GMM improves performance, but the lack of CRP leads to results inferior to those of GMDI, demonstrating the importance of both GMM modeling and the use of CRP. Please let us know if we need to provide further explanation. Thank you very much.

---

> > ### Comment · Area_Chair_HVnC · 2024-08-11
> >
> > Dear Reviewer uzxH,
> >
> > Just wanted to check if the authors' further response addressed your remaining concerns. Feel free to ask follow-up questions if needed. Thanks!
> >
> > Best,
> >
> > AC

---

> ### Author Response · Authors · 2024-08-14
> **Thanks for the prompt response.**
>
> Thank you for your detailed comments. The following is our responses to the concerns mentioned in the comments.
>
> **1. "However, the study primarily builds upon the existing framework of VDI, and its theoretical developments, which are closely aligned with those of NDI, are mostly relegated to the appendix without appropriate citations. This lack of clarity suggests that some key proofs might be incorrectly attributed to this work."**
>
> Thank you for pointing out this issue. Our theorems/lemmas are indeed partially based on VDI, and we will carefully review them to ensure that VDI is correctly cited in both the theorems/lemmas in the main body of our paper and the proofs in the appendix in the final version.
>
> Compared to VDI, the theoretical contribution of the proposed GMDI is that it proves GMDI achieves a more stringent evidence lower bound, as well as a tighter upper bound on the objective. Theoretically, it demonstrates that GMDI can effectively infer optimal domain indices.
>
> **2.  "Additionally, the use of the Chinese Restaurant Process (CRP) to manage an unknown number of Gaussian components is not innovative, as thoroughly discussed in prior research [1]. Therefore, both the technical novelty and theoretical contributions of this paper seem incremental."**
>
> Thanks for pointing out this. The main innovation of our proposed GMDI is that, to the best of our knowledge, GMDI is the first to introduce a CRP-based Gaussian mixture model to represent the domain index.
>
> **3. "The experiments presented are not sufficiently comprehensive. I can understand that the authors have difficulties in conducting experiments using additional GMM variants, yet such comparisons are essential for a robust evaluation. Notably in the experiments, the CRP model frequently identifies the optimal number of components as two or three. It is important to figure out why CRP outperforms fixed-component GMMs, given that the "optimal" number of Gaussian components is known. A more thorough ablation study could clarify these results."**
>
> Thanks for mentioning this. Since fixed-component GMMs are fundamentally similar, none of them can adaptively determine the number of mixture components. It is neither realistic nor meaningful to compare all variants of fixed-component GMMs within a short period. We have already conducted a detailed comparison with the Gumbel softmax version (i.e., GMDI w/o CRP). The experimental results show that our proposed GMDI demonstrates clear advantages. We leave the extension of comparative experiments with additional GMM variants for future research.
>
> Regarding why the CRP model (i.e., GMDI) outperforms the fixed-component GMM (i.e., GMDI w/o CRP), it may be due to the complexities of the training process and parameter search space. Firstly, during training, the varying sample distribution within each batch can make it difficult for the fixed-component GMM (i.e., GMDI w/o CRP) to adapt well under a fixed K, potentially leading to suboptimal performance and convergence challenges. In contrast, the CRP model (i.e., GMDI) can automatically adapt to the sample distribution during training, and experiments have observed that GMDI converges faster to better performance, which aligns with our expectations.
>
> Secondly, due to the complex parameter space, it is challenging to fine-tune both the fixed-component GMM (i.e., GMDI w/o CRP) and GMDI to their optimal performance within a short time frame. However, under fair comparison conditions, GMDI demonstrates superior performance and convergence speed compared to the fixed-component GMM (i.e., GMDI w/o CRP) within a limited search space, while significantly reducing the cost of parameter searching. As shown in the experimental results in the table below, we have tested various hyperparameters, K and $\tau$, as thoroughly as possible. The experiments indicate that even under the fairest possible comparison conditions, the fixed-component GMM (i.e., GMDI w/o CRP) struggles to achieve better performance, thereby validating our conclusions.
>
> （1）GMDI w/o CRP on *CompCars* dataset:
>
> | Accuracy(%) | K=1  | K=2  | K=3      | K=4  | K=5  |
> | ----------- | ---- | ---- | -------- | ---- | ---- |
> | $\tau$=100  | 42.6 | 42.1 | 42.4     | 41.8 | 40.0 |
> | $\tau$=50   | 42.6 | 42.8 | **43.0** | 42.1 | 39.8 |
> | $\tau$=10   | 42.6 | 41.8 | 42.0     | 42.9 | 42.4 |
> | $\tau$=1    | 42.6 | 42.7 | 36.9     | 30.3 | 30.3 |
> | $\tau$=0.1  | 42.6 | 38.5 | 41.8     | 30.4 | 30.3 |

---

> > ### Author Response · Authors · 2024-08-14
> > **Thanks for the prompt response.**
> >
> > **4. "Furthermore, the methodology should be tested on data featuring more complex multimodal distributions, which may require a larger number of Gaussian components for accurate representation of the index. Such evaluation would help validate the method’s effectiveness and demonstrate more advantages over a fixed components method."**
> >
> > Thank you for bringing this to our attention. We have already conducted comparative experiments on the most complex dataset used in VDI, *CompCars*, and the experimental results show that our proposed GMDI demonstrates clear advantages. Due to the scarcity of datasets with ground-truth domain indices, we leave the extension of comparative experiments on new datasets for future research.

---

### Author Rebuttal · Authors · 2024-08-07

We thank all the respected reviewers for their detailed comments and believe that all the mentioned issues can be properly addressed in the final version of our paper. The major concerns lie in the ablation and computational cost experiments. We take this opportunity to clarify these issues and present our responses accordingly. Regarding the ablation and computational cost experiments, we have followed the suggestions and provided detailed comparison results in the following Ablation Study section.

We appreciate the reviewers for highlighting the issue of computational cost associated with the Chinese Restaurant Process (CRP). The CRP is indeed computationally intensive. To improve computational efficiency, the proposed GMDI employs the stick-breaking construction of the CRP by specifying an upper bound $K$ for the number of components in the Gaussian mixture. By selecting an appropriate $K$, we are able to leverage the benefits of the CRP while reducing computational cost. More details are available in Appendix A of the paper.

To evaluate the impact and computational cost of the CRP, we conduct ablation and computational cost experiments. Following the reviewers' suggestions, we implement GMDI w/o CRP using Gumbel softmax. The number of components for GMDI w/o CRP is set to the upper bound $K$ of GMDI, and the hyperparameter temperature $\tau$ for Gumbel softmax ranges from 0.1 to 50 (with the best performance reported). "Total time" refers to the total training duration, which concludes when the loss converges. The experimental results on three datasets are shown in the following tables.

(1) *TPT-48*(W->E) dataset:

| Method       | MSE       | Total time | Epochs | Time per epoch |
| ------------ | --------- | ---------- | ------ | -------------- |
| VDI          | 2.496     | 1h 24m 18s | 400    | 13s            |
| GMDI w/o CRP | 2.471     | 2h 14m 54s | 500    | 16s            |
| GMDI         | **2.087** | 1h 31m 13s | 300    | 18s            |

(2) *TPT-48*(N->S) dataset:

| Method       | MSE       | Total time | Epochs | Time per epoch |
| ------------ | --------- | ---------- | ------ | -------------- |
| VDI          | 3.160     | 1h 51m 43s | 500    | 13s            |
| GMDI w/o CRP | 3.050     | 2h 22m 58s | 500    | 17s            |
| GMDI         | **2.493** | 2h 2m 35s  | 400    | 18s            |

(3) *CompCars* dataset:

| Method       | Accuracy(%) | Total time | Epochs | Time per epoch |
| ------------ | ----------- | ---------- | ------ | -------------- |
| VDI          | 42.5        | 3h 13m 15s | 600    | 19s            |
| GMDI w/o CRP | 43.0        | 4h 3m 48s  | 700    | 21s            |
| GMDI         | **44.4**    | 4h 16m 14s | 600    | 26s            |

We find that although the proposed GMDI has a longer "Time per epoch" compared to GMDI w/o CRP, it converges faster due to the flexible number of components adaptively controlled by CRP. Therefore, the "Total time" is roughly the same as GMDI w/o CRP. On the *TPT-48*(W->E) dataset, due to faster convergence, the "Total time" of GMDI is significantly less than that of GMDI w/o CRP and is even comparable to VDI. In all three datasets, the performance of GMDI w/o CRP is significantly worse than that of GMDI. On the two *TPT-48* datasets, compared to the baseline VDI, GMDI w/o CRP reduces MSE by only 1% and 3%, whereas GMDI reduces MSE by 16% and 21%, far surpassing GMDI w/o CRP. On the *CompCars* dataset, GMDI's accuracy significantly higher than that of GMDI w/o CRP. These results indicate that although using a fixed-component GMM is simpler, the computational costs are roughly equivalent to using CRP, but the performance is far inferior to CRP, demonstrating the significance of CRP in GMDI.

Additionally, compared to VDI, which models the domain index as a single Gaussian distribution, GMDI's computational costs are only slightly higher, yet its performance is far superior. For large-scale datasets with numerous domains, modeling the domain index as a simple single Gaussian distribution may result in poor performance due to the dataset's complexity. The experimental results indicate that GMDI has broad applicability.

---

### Decision · Program_Chairs · 2024-09-25

**Decision:**

Accept (poster)

**Comment:**

The paper builds upon Variational Domain Indexing (VDI) from ICLR 2023 to introduce the Gaussian Mixture Domain-Indexing (GMDI) algorithm to tackle the challenge of domain adaptation when domain indices are not available. Unlike traditional approaches that assume a single Gaussian prior for domain indices, GMDI models these indices using a Gaussian Mixture Model, with the number of components dynamically determined by a Chinese Restaurant Process. This approach is intended to provide more flexibility and better adaptation to diverse domains.

### Strengths

- GMDI is innovative in its use of a Gaussian Mixture Model to capture the inherent structures among different domains, which allows for a more flexible and powerful inference of domain indices.
- The method is theoretically well-founded, with proofs that demonstrate a more stringent evidence lower bound and a tighter upper bound compared to existing methods.
- Extensive experiments show that GMDI outperforms state-of-the-art methods in both classification and regression tasks, achieving significant improvements in accuracy and mean squared error (MSE).

### Weaknesses

- The model’s learning loss is complex, involving multiple conditional Kullback-Leibler divergences, which could make implementation and interpretation challenging.
- The use of the Chinese Restaurant Process and Gaussian Mixture Model increases computational overhead, potentially limiting the scalability of the method for large-scale datasets or real-time applications.
- The paper lacks clarity in some areas, such as the definitions of 'local' and 'global' domain indices, and certain equations may contain typos, which could confuse readers.

Some concerns have been addressed by the authors during the rebuttal period.

Near the end of the discussion, there are new concerns on the citation of VDI on the theorems, raised by the reviewers. As prompted by the reviewers, I also briefly checked the paper myself. I think these concerns are kind of valid. For example, most theorems/lemmas in the paper are from the VDI paper, and therefore should be proper attributed to VDI by citation. Another example is Line 255, where "VDI" should be "GMDI"? That said, I do agree with the majority that the paper has its technical merit. What is more important at this point is proper attribution and citation, especially on the theory part. In my opinion, properly attributing the theories will not affect the value of GMDI; on the contrary, it enhances GMDI's credibility and rigor. The authors did reply to me, promising that they will address these concerns fully in the revision.

Below I summarize several points of revision suggestions after I went over GMDI and VDI:

+ Fix the typo in Line 255 from VDI to GMDI
+ **Lemma 1 and Lemma 4.1 in VDI:** After stating Lemma 1, **cite VDI** and **clearly state** that `“the main difference between and Lemma 1 in GMDI and Lemma 4.1 in VDI [37] is the last two KL terms and the inclusion of $v$”`.
+ **Lemma 2 and Lemma 4.2 in VDI:** Lemma 2 is exactly the same as Lemma 4.2 in VDI. Therefore, please properly **cite VDI inside the bracket of Lemma 2’s title, i.e., `“(Information Decomposition of the Adversarial Loss **[37]**)”`.
+ **Theorem 1 and Theorem 4.1 in VDI:** After stating Theorem 1, **cite VDI** and **clearly state** that `“the main difference between VDI and GMDI is the inclusion of $v$"`.
+ **Theorem 2 and Theorem 4.2 in VDI:** After stating Theorem 2, **cite VDI** and **summarize** the main difference between VDI and GMDI. There are more differences in this theorem; I believe it would give more insight to readers on the contribution of GMDI beyond VDI if the authors can clarify them well here.

I agree with the majority that this is an interesting paper, with solid extension to VDI, which itself is a promising new direction in domain adaptation and (hierarchical) Bayesian deep learning.

I also tend to trust the authors’ promise on the revision and proper citation/attribution and will give a *conditional accept*. Note that
+ I will help shepherd the revision, and make sure all points, especially those I listed above, are fully incorporated in the revision. Since this is a matter of academic rigor; otherwise this paper would have the risk of plagiarism.
+ While I recommend acceptance for now, I will stay in touch with the PC Chairs and SAC, and reserve the right to retract the paper if the camera ready does not fully address the concerns above.